# Quantitative definition of neurobehavior, vision, hearing and brain volumes in macaques congenitally exposed to Zika virus

**Michelle R. Koenig[1,2], Elaina Razo[3], Ann Mitzey[2], Christina M. Newman[1], Dawn M. Dudley[1], Meghan E. Breitbach[1], Matthew R. Semler[1], Laurel M. Stewart[1], Andrea M. Weiler[4], Sierra Rybarczyk[4], Kathryn M. Bach[5], Mariel S. Mohns[1], Heather A. Simmons[4], Andres Mejia[4], Michael Fritsch[1], Maria Dennis[6], Leandro B. C. Teixeira[7], Michele L. Schotzko[4], T. Michael Nork[8], Carol A. Rasmussen[8], Alex Katz[8], Veena Nair[9], Jiancheng Hou[9], Amy Hartman[10], James Ver Hoeve[8], Charlene Kim[8], Mary L. Schneider[5], Karla Ausderau[5], Sarah Kohn[9], Anna S. Jaeger[11], Matthew T. Aliota[11¤], Jennifer M. Hayes[4], Nancy Schultz-Darken[4], Jens Eickhoff[12], Kathleen M. Antony[13], Kevin Noguchi[14], Xiankun Zeng[15], Sallie Permar[6], Vivek Prabhakaran[9], Saverio Capuano, III[4], Thomas C. Friedrich[4,7], Thaddeus G. Golos[2,4,13], David H. O'Connor[1,4], Emma L. Mohr[3]***

1 Department of Pathology and Laboratory Medicine, UW-Madison, Madison, Wisconsin, United States of America, 2 Department of Comparative Biosciences, UW-Madison, Madison, Wisconsin, United States of America, 3 Department of Pediatrics, UW-Madison, Madison, Wisconsin, United States of America, 4 Wisconsin National Primate Research Center, UW-Madison, Madison, Wisconsin, United States of America, 5 Department of Kinesiology, UW-Madison, Madison, Wisconsin, United States of America, 6 Department of Pediatrics, Duke University, Durham, North Carolina, United States of America, 7 Department of Pathobiological Sciences, UW-Madison, Madison, Wisconsin, United States of America, 8 Department of Ophthalmology and Visual Sciences, UW-Madison, Madison, Wisconsin, United States of America, 9 Department of Radiology, UW-Madison, Madison, Wisconsin, United States of America, 10 Department of Communication Sciences and Disorders, UW-Madison, Madison, Wisconsin, United States of America, 11 Department of Veterinary and Biomedical Sciences, University of Minnesota, Minneapolis, Minnesota, United States of America, 12 Biostatistics and Medical Informatics, UW-Madison, Madison, Wisconsin, United States of America, 13 Department of Obstetrics and Gynecology, UW-Madison, Madison, Wisconsin, United States of America, 14 Department of Psychiatry, Washington University School of Medicine, Saint Louis, Missouri, United States of America, 15 United States Army Medical Research Institute of Infectious Diseases, Fort Detrick, Frederick, Maryland, United States of America

¤ Current address: Department of Veterinary and Biomedical Sciences, University of Minnesota, Saint Paul, Minnesota, United States of America

* emohr2@wisc.edu

**Data Availability Statement:** Data are available: (https://go.wisc.edu/0n9g6o).

## Abstract

Congenital Zika virus (ZIKV) exposure results in a spectrum of disease ranging from severe birth defects to delayed onset neurodevelopmental deficits. ZIKV-related neuropathogenesis, predictors of birth defects, and neurodevelopmental deficits are not well defined in people. Here we assess the methodological and statistical feasibility of a congenital ZIKV exposure macaque model for identifying infant neurobehavior and brain abnormalities that may underlie neurodevelopmental deficits. We inoculated five pregnant macaques with ZIKV and mock-inoculated one macaque in the first trimester. Following birth, growth, ocular structure/function, brain structure, hearing, histopathology, and neurobehavior were quantitatively assessed during the first week of life. We identified the typical pregnancy outcomes of congenital ZIKV infection, with fetal demise and placental abnormalities. We estimated sample sizes needed to define differences between groups and demonstrated that future

**Funding:** This study was supported by the National Institutes of Health (P01 AI132132 (DHO), R01 AI138647 (DHO), R01 AI116382-01A1S1 (DHO), K08 AI139341 (ELM), R01 AI132563 (MTA and TCF)) and the Pediatric Infectious Diseases Society funded by Stanley A. Plotkin and Sanofi Pasteur (www.pids.org) (ELM). This study was also supported by the Clinical and Translational Science Award (CTSA) program, through the NIH National Center for Advancing Translational Sciences (NCATS), grant UL1TR002373-02 and TL1002375-02 (ELM). This study was supported in part by the Core Grant for Vision Research from the NIH to the University of Wisconsin-Madison (P30 EY016665) (PI: Curtis Brandt), McPherson Eye Research Institute's Retina Research Foundation Kathryn & Latimer Murfee Chair (TMN), and an unrestricted grant from Research to Prevent Blindness to the University of Wisconsin Department of Ophthalmology and Visual Sciences (TMN). The content is solely the responsibility of the authors and does not necessarily represent the official views of the NIH.

**Competing interests:** DHO is a paid consultant for Battelle, devoted to research in the areas of assisting in the design and interpretation of their nonhuman primate ZIKV studies. His relationship does not carry with it any restrictions on publication, and any associated intellectual property will be disclosed and processed according to UW-Madison policy. None of the animals used in this study are involved in any studies with Battelle.

studies quantifying brain region volumes, retinal structure, hearing, and visual pathway function require a sample size of 14 animals per group (14 ZIKV, 14 control) to detect statistically significant differences in at least half of the infant exam parameters. Establishing the parameters for future studies of neurodevelopmental outcomes following congenital ZIKV exposure in macaques is essential for robust and rigorous experimental design.

## Introduction

A spectrum of abnormalities result from in utero Zika virus (ZIKV) exposure and includes birth defects, termed congenital Zika syndrome, and neurodevelopmental deficits. Approximately 10% of infants have defects apparent at birth, including ocular anomalies, brain anomalies, cranial dysmorphologies, congenital contractures and hearing loss [1]. Twenty-eight percent of ZIKV-exposed children are asymptomatic at birth and present with neurodevelopmental deficits in early childhood [1–4]. These neurodevelopmental abnormalities include delays in gross motor, fine motor, and problem-solving skills [3, 5, 6] along with diminished mobility, communication, and social cognition [4, 5]. There are currently no tools to distinguish the infants that will develop neurodevelopmental deficiencies from those who will remain asymptomatic, as the long-term outcomes in children who were born during the 2015–2016 ZIKV epidemic remain to be fully defined [3, 4, 6–8].

Translational models of ZIKV infection during pregnancy and subsequent infant neurodevelopment are necessary to define the neuropathogenesis and early neural predictors of deficits. Different fetal exposure times during gestation, ZIKV strains, and genetic and socioeconomic backgrounds all confound our ability to understand the neuropathogenesis of in utero ZIKV infection in human studies. Furthermore, in human studies, we are unable to correlate histopathology with functional outcomes. Therefore, translational animal models must be used to define neuropathogenesis and predictors of deficits, with the end goal of identifying targets for intervention and therapy [9–11].

Currently, there is no established nonhuman primate model for defining the neuropathogenesis of the most common phenotype of congenital ZIKV infection: children who develop neurodevelopmental deficits but lack the birth defects found in congenital Zika syndrome [12]. Early macaque studies have provided insight into in utero ZIKV infection and defined fetal neuropathology of gestational ZIKV infection [11, 13, 14], placental pathology [13], fetal tissue viral distribution [14–17], and birth defects in neonates [18]. None of these studies have defined long-term neurodevelopmental deficits or outlined a clear study design for how to evaluate long-term neurodevelopment in congenital ZIKV-exposed infant macaques. Such a study design needs to be well planned, with proven quantitative neurodevelopmental outcomes and sufficient sample sizes.

Before long-term studies defining the pathogenesis of neurodevelopmental deficits in ZIKV-exposed macaques are undertaken, we must determine whether tests defining neurodevelopmental outcomes, such as quantitative structural brain imaging, ocular examinations and hearing tests, are feasible in infant macaques. This feasibility examination includes translating appropriate human quantitative clinical exams to an infant macaque population and determining what sample sizes are required for statistically defining differences between control and ZIKV-exposed infants. These short-term macaque feasibility studies would be analogous to what is required in human clinical trial work [19, 20].

This study aims to demonstrate the feasibility of quantitatively defining developmental outcomes and viral tissue tropism in a macaque model of congenital ZIKV exposure by

concentrating infant exams and tissue viral RNA (vRNA) analyses in the first week of life. We performed the same qualitative clinical exams that human infants receive to assess for birth defects [21]. We developed a panel of quantitative infant exams to define subtle abnormalities in structural brain volumes, visual pathway structure and function, hearing and neurobehavior. We defined the number of infants needed detect significant differences in these quantitative infant exam parameters in post-hoc power analyses. Tissue viral loads and histopathology were also assessed in the infant macaques. Our results demonstrate that ZIKV vRNA is not identified in all ZIKV-exposed infants and it is feasible to quantitatively define infant neurodevelopment with moderately sized studies.

## Materials and methods

### Study design

Indian-origin rhesus macaques (*Macaca mulatta*) were inoculated with ZIKV alone or phosphate buffered saline (PBS) alone during the first trimester (term is 165±10 days) (Table 1). All dams were part of the Specific Pathogen Free (SPF) colony at the Wisconsin National Primate Research Center (WNPRC) and were free of *Macacine alphaherpesvirus 1* (Herpes B), simian retrovirus type D (SRV), simian T-lymphotropic virus type 1 (STLV), and simian immunodeficiency virus (SIV).

### Ethics statement

All monkeys are cared for by the staff at the WNPRC in accordance with the regulations and guidelines outlined in the Animal Welfare Act and the Guide for the Care and Use of Laboratory Animals, the recommendations of the Weatherall report (https://royalsociety.org/topics-policy/publications/2006/weatherall-report), and the principles described in the National Research Council's Guide for the Care and Use of Laboratory Animals. The University of Wisconsin—Madison Institutional Biosafety Committee approved this work under protocol number B00000117. See study approval section below for animal protocol details.

### Care & use of macaques

All animals were housed in enclosures with required floor space and fed using a nutritional plan based on recommendations published by the National Research Council. Dams were fed a fixed formula, extruded dry diet with adequate carbohydrate, energy, fat, fiber, mineral, protein, and vitamin content. Macaque dry diets were supplemented with fruits, vegetables, and other edible objects (e.g., nuts, cereals, seed mixtures, yogurt, peanut butter, popcorn, marshmallows, etc.) to provide variety to the diet and to inspire species-specific behaviors such as foraging. Infants were fed 5% dextrose for the first 24 hours of life and liquid formula subsequently. To further promote psychological well-being, animals were provided with food

**Table 1. Characteristics of the dams inoculated with ZIKV or PBS.**

| | Dam identification numbers | | | | | |
|---|---|---|---|---|---|---|
| | ZIKV inoculation | | | | | PBS* inoculation |
| | 664184 | 484880 | 795784 | 918724 | 730267 | 020101 |
| Maternal age at conception (years) | 4 | 11 | 16 | 14 | 13 | 8 |
| Number of prior pregnancies | 0 | 5 | 4 | 7 | 6 | 3 |
| Gestational age at ZIKV inoculation (days) | 46 | 43 | 41 | 50 | 44 | 47 |

*Phosphate buffered saline (PBS).

enrichment, structural enrichment, and/or manipulanda. Environmental enrichment objects were selected to minimize chances of pathogen transmission from one animal to another and from animals to care staff. While on study, all animals were evaluated by trained animal care staff at least twice each day for signs of pain, distress, and illness by observing appetite, stool quality, activity level, and physical condition. Animals exhibiting abnormal presentation for any of these clinical parameters were provided appropriate care by certified veterinarians. Prior to all minor/brief experimental procedures, macaques were sedated using ketamine anesthesia and monitored regularly until fully recovered from sedation.

For breeding, the female macaques were co-housed with a compatible male and observed daily for menses and breeding. Pregnancy was detected by ultrasound examination of the uterus at approximately 20–24 gestation days (gd) following the predicted day of ovulation. The gd was estimated (+/- 2 days) based on the dam's menstrual cycle, observation of copulation, and the greatest length of the fetus at initial ultrasound examination which was compared to normative growth data in this species [22]. For physical examinations, virus inoculations, some ultrasound examinations, and blood and swab collections, the dam was anesthetized with an intramuscular dose of ketamine (10 mg/kg). Blood samples from the femoral or saphenous vein were obtained using a vacutainer system or needle and syringe. Pregnant macaques were monitored daily prior to and after viral inoculation for any clinical signs of infection (e.g., diarrhea, inappetence, inactivity, fever and atypical behaviors).

## Inoculation and monitoring

Macaques were inoculated subcutaneously with $1x10^4$ PFU Zika virus/*H.sapiens-tc/PUR/2015/PRVABC59_v3c2* (PRVABC59, GenBank: KU501215). This virus was originally isolated from a traveler to Puerto Rico and passaged three times on Vero cells (American Type Culture Collection (ATCC): CCL-81). The seed stock was obtained from Brandy Russell (CDC, Ft. Collins, CO). Virus stocks were prepared by inoculation onto a confluent monolayer of C6/36 cells (*Aedes albopictus* mosquito larval cells; ATCC: CCL-1660) with two rounds of amplification. The animals were anesthetized as described above, and 1 mL of inoculum at $1 \times 10^4$ PFU dilution in PBS was administered subcutaneously over the cranial dorsum. Post-inoculation, the animals were closely monitored by veterinary and animal care staff for adverse reactions or any signs of disease.

## Pregnancy monitoring and fetal measurements

Weekly ultrasounds were conducted to observe the health of the fetus and to obtain measurements including fetal femur length (FL), biparietal diameter (BPD), head circumference (HC), abdominal circumference (AC) and heart rate as previously described [16]. Growth curves were developed for FL, BPD, and HC using mean measurements and standard deviations from Tarantal et al. [22] and the ultrasound measurements were plotted against this normative data. Doppler ultrasounds to measure fetal heart rate were performed biweekly.

## vRNA isolation from body fluids and tissues and qRT-PCR

RNA was extracted from 300 μl of plasma using the Viral Total Nucleic Acid Purification kit (Promega, Madison, WI, USA) on a Maxwell 16 MDx instrument. qRT-PCR was performed as previously described [23]. The limit of quantification for the assay is 100 copies/mL for qRT-PCR from plasma. Fetal and maternal-fetal interface tissues were preserved with RNAlater® (Invitrogen, Carlsbad, CA) immediately following collection. RNA was isolated from maternal and fetal tissues using the Trizol Plus RNA Purification kit (Invitrogen, Carlsbad, CA) following the manufacturer's instructions or a similar method described by Hansen et al.

[24]. In the latter method, RNA was recovered from up to 200 mg of tissue disrupted in TRIzol (Life Technologies, Waltham, MA) with 2 x 5 mm stainless steel beads using the TissueLyser (Qiagen, Hilden, Germany) for 3 minutes at 25 r/s twice. Following homogenization, samples in TRIzol were separated using Bromo-chloro-propane (Sigma). The aqueous phase was collected and glycogen was added as a carrier. The samples were precipitated in isopropanol and washed in 70% ethanol. RNA was fully re-suspended in 5 mM tris pH 8.0.

## Plaque Reduction Neutralization Test (PRNT)

Macaque serum samples were screened for ZIKV neutralizing antibodies using a plaque reduction neutralization test (PRNT). Endpoint titrations of reactive sera, using a 90% cutoff (PRNT90), were performed as described [25] against ZIKV strain PRVABC59. Briefly, ZIKV was mixed with serial 2-fold dilutions of serum for 1 hour at 37˚C prior to being added to Vero cells and neutralization curves were generated using GraphPad Prism software (La Jolla, CA). The resulting data were analyzed by nonlinear regression to estimate the dilution of serum required to inhibit 90% of infection.

## Whole virion ZIKV-specific binding antibody ELISA

High-binding 96-well ELISA plates (Greiner; Monroe, NC) were coated with 40 ng/well of 4G2 monoclonal antibody, which was produced in a mouse hybridoma cell line (D1-4G2-4-15, ATCC; Manassas, VA), diluted to 0.8 ng/uL in 0.1M carbonate buffer (pH 9.6) and incubated overnight at 4˚C. Plates were blocked with 1X Tris-buffered saline containing 0.05% Tween-20 and 5% normal goat serum (cat.# G6767, Sigma-Aldrich, St. Louis, MO) or 1 hour at 37˚C, followed by incubation with $1.15 \times 10^5$ focus-forming units (ffu)/well Zika virus (PRVABC59, BEI; Manassas, VA) for 1 hour at 37˚C. Serum samples were tested at a dilution of 1:12.5–204,800 in serial 4-fold dilutions and incubated for 1 hour at 37˚C, along with a ZIKV-specific monoclonal antibody, H24 (10 ug/mL), isolated from a ZIKV-infected rhesus macaque. Horseradish peroxidase (HRP)-conjugated mouse anti-monkey IgG secondary antibody (Southern BioTech; Birmingham, AL) was used at a 1:4,000 dilution and incubated at 37˚C for 1 hour, followed by the addition of SureBlue Reserve TMB Substrate (KPL; Gaithersburg, MD). Reactions were terminated by Stop Solution (KPL; Gaithersburg, MD) after a 7-minute incubation per plate in the dark. Optical density (OD) was detected at 450 nm on a Victor X Multilabel plate reader (PerkinElmer; Waltham, MA). Binding was considered detectable if the sample OD value at the lowest dilution was greater than that of the background OD, defined as the OD value of the negative control at the lowest dilution plus 2 standard deviations (SD). For samples considered positive, their OD values for the serial dilution were entered into Prism v8 (GraphPad Software; San Diego, CA) to determine the 50% effective dilution ($ED_{50}$). Briefly, the $ED_{50}$ was calculated by transforming the fold dilution into $log_{10}$. The transformed data was then analyzed using a sigmoidal dose-response nonlinear regression model. Any sample considered negative was assigned an $ED_{50}$ of 12.5, the lowest dilution tested, because $ED_{50}$ cannot be accurately calculated below the lowest dilution tested. The $log_{10}$ 50% effective dilutions ($ED_{50}$) were calculated for IgG binding responses against the whole virion and compared between 0, 7, 14 and 28 dpi time points.

## ZIKV IgM enzyme-linked immunosorbent assay (ELISA)

Infant macaque serum samples were screened for anti-ZIKV IgM antibodies using commercial human anti-ZIKV IgM ELISA assays targeting ZIKV nonstructural protein 1 (NS1) antigen. We utilized a sandwich ELISA (cat# ab213327, Abcam, Cambridge, Massachusetts, USA) and an indirect ELISA (cat# el26689601M, Euroimmun, Mountain Lakes, New Jersey, USA),

following manufacturer instructions. Controls included manufacturer-provided human serum samples and rhesus macaque serum collected at 0 and 14 days post-infection with PRVABC59 from an adult rhesus macaque (animal ID 244667). Infant serum samples were run in triplicate. All samples and controls were read on a Synergy HTX plate reader at 450 nm. Sample values were reported in "Abcam units" (Abcam) or "ratio" (Euroimmun), which are the ratio of sample serum absorbance to the absorbance of a manufacturer-provided solution containing the upper limit of the reference range of non-infected biological sample ("cutoff" (Abcam) or "calibrator" (Euroimmun)). This method of reporting results is recommended by the manufacturers to control for inter-assay variability.

## Cesarean delivery and maternal necropsy

All infants were delivered by cesarean section at approximately 155 gestational days (gd), except for the single fetal demise at 133 gd. Infants were delivered approximately 10 days before the typical gestational age of a natural birth at the WNPRC (166 gd) to ensure that the placenta could be collected for evaluation. Amniotic fluid was collected just prior to infant delivery via aspiration with a syringe and needle inserted through the chorioamniotic membranes. Sterile instruments were used for the dissection and collection of all maternal and maternal-fetal interface tissues during the gross post-mortem examination. Each tissue was collected with a unique set of sterile instruments and placed in a separate sterile petri dish before transfer to RNAlater for ZIKV qRT-PCR or fixed for histology to prevent cross-contamination.

## Infant care

After delivery, infants were dried, stimulated and received respiratory support as necessary, and placed in a warmed incubator. All liveborn infants were transferred to the nursery where they remained until euthanasia and necropsy on day of life 7–8. Infants were reared in the nursery to enable continuous access to the infants for testing and to prevent confounding of maternal rearing differences on neurobehavior. During this period, multiple examinations were completed as described in Fig 1. Infant weights from day of life (DOL) 0 to DOL 8 were recorded daily by the nursery staff. The volumes of 5% dextrose and liquid formula consumed by each infant were recorded.

## Neurobehavioral assessments

We evaluated neonatal macaque neurobehavior with a well-validated assessment developed for rhesus macaques less than 1 month of age, termed the Schneider Neonatal Assessment for Primates (SNAP) [26–30], which is based on the Brazelton Newborn Behavioral Assessment Scale [31]. The Schneider Neonatal Assessment for Primates (SNAP) [29], a 20 minute battery of developmental tests, was administered at 1, 3–5, and 6–7 DOL, with the day of birth considered DOL 0 (summarized in S1 Table). This assessment captures detailed developmental behaviors in macaques less than 1 month of age. The SNAP assesses neonatal neurobehavior using 68 individually scored test items in four constructs (i.e. domains): motor maturity and activity, orientation, sensory, and state control (S2 Table). Testing occurred midway between feedings at approximately the same time each day. Ratings were based on a five-point Likert scale ranging from 0 to 2, with a higher value representing higher performance. An average score was calculated for each of the four constructs. Examiners were trained in standardized administration and scoring procedures by the SNAP developer, M. Schneider, requiring a check-out protocol prior to administration. Two examiners (M. Schneider and K. Ausderau) performed the neurobehavioral testing and scoring to ensure test administration reliability (>95%).

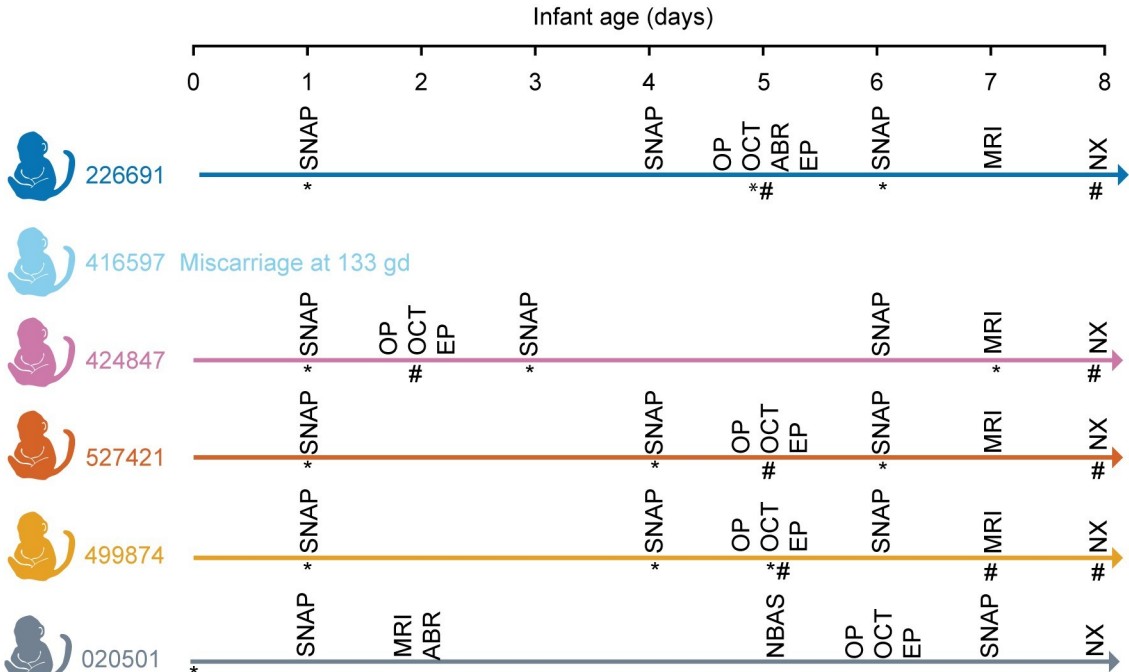

**Fig 1. Infant examination schedule.** Infants were evaluated by the Schneider Neonatal Assessment for Primates (SNAP), ophthalmic exam (OP), optical coherence tomography (OCT), auditory brainstem response (ABR), visual electrophysiology (EP) and brain MRI prior to euthanasia and necropsy (NX). Urine (*) and blood (#) were collected at the indicated times. Symbols are under the collection line for an individual animal.

## Ophthalmic exam

Infants were anesthetized as described in S3 Table for eye exams performed by a human ophthalmologist with retinal fellowship training (M. Nork). Pupillary reactivity was assessed, and intraocular pressures were obtained prior to eye dilation with ophthalmic drops (0.05% proparacaine, tropicamide, phenylephrine). Slit-lamp biomicroscopy and indirect ophthalmoscopy were performed after pupillary dilation.

## Optical coherence tomography

Spectral-domain optical coherence tomography (SD-OCT) uses non-invasive light waves to evaluate the anterior segment of the eye (i.e. the ocular structures anterior to the vitreous) and the retina. Scans of the retina and anterior segment were carried out in both eyes of most infants using a Heidelberg™ Spectralis HRA + OCT (Heidelberg™ Engineering, Heidelberg, Germany) instrument. Segmentation and determination of corneal and retinal layer thickness was performed using combined manual and automatic segmentation algorithms from both Heidelberg and EXCELSIOR [32]. EXCELSIOR Preclinical's functionalities for retinal segmentation (EdgeSelect™) and automated image analysis were used to calculate mean thicknesses for retinal layers.

## Visual electrophysiology

To objectively evaluate visual function in ZIKV-exposed neonatal rhesus macaques, a clinically trained visual electrophysiologist (J. Ver Hoeve) performed standard visual electrodiagnostic procedures including a full-field electroretinogram (ERG) and the cortical-derived visual

evoked potential (VEP) [33]. The light-adapted (LA) full-field flash ERG, recorded on a rod-saturating background, measures the electrical activity generated predominantly by cone photoreceptor and bipolar cells, which are found in high density in the primate macula and are primarily responsible for light-adapted, high acuity and color vision. The ERG is used clinically to assess generalized retinal function under light-adapted (focused on cone photoreceptors) and dark-adapted (focused on rod photoreceptors) conditions [33] and has been used to characterize maculopathy in acute ZIKV infection [34]. The VEP reflects the function of the entire visual pathway from the retina via the optic nerve to the visual cortex of the brain [33]. We performed standard LA ERGs to measure how the cone photoreceptors function.

Photopic full-field ERGs to a series of flash strengths and photopic flash visual-evoked potentials were recorded in that order. Measurements were recorded using a BigShot™ electro-diagnostic system (LKC Technologies™, Gaithersburg, MD). When isoflurane was used during an earlier procedure in the same sedation event, a washout period was allowed before visual electrophysiology studies to minimize isoflurane suppression of cortical activity [35]. Corneas were anesthetized with topical 0.5% proparacaine prior to application of ERG-149 jet™ (Universo™, Switzerland) contact lens electrodes and a conductive wetting solution. Reference electrodes were subdermal stainless-steel needle electrodes inserted near the ipsilateral outer canthus of each eye. Visual evoked potentials were recorded from two active subdermal electrodes situated approximately 1 cm superior to the occipital ridge and 1 cm lateral to the midline; VEP reference electrodes were situated adjacent to one another along the midline at the vertex. Replicates (2–4) of 80 flashes each were performed on each macaque. ERG and VEP waveforms were processed off-line and machine scored using software written in Matlab™ (Nattick, MA). Photopic flash VEPs were quantified as the root-mean-square of the response from 50–150 msec post flash. S1 Fig illustrates the waveform characteristics quantified with the ERGs and VEPs.

### Auditory brainstem response testing

Hearing, or auditory brainstem function, was assessed with auditory brainstem response audiometry, which measures brainstem evoked potentials generated by a brief click. Examinations were performed by a human audiologist with pediatric training (A. Hartman). Auditory brainstem response (ABR) thresholds were obtained for auditory (click) stimuli, as described in [36], using the Biologic Navigator Pro system. Ambient noise level was minimized. For the ABR, needle electrodes were placed at the brow ridge (positive input) and behind the right pinna (negative input) for channel 1 and from the brow ridge (positive input) and behind the left pinna (negative input) for channel 2. An electrode was placed below the brow ridge on the forehead for the ground. Electrode impedances were below 10 kohm for all electrodes. Physiological filters were set to pass 100–3000 Hz. The stimuli were clicks with rarefaction and condensation alternating polarity and a Blackman window with 2-ms rise-fall and 1 ms plateau times. Insert earphones (Etymotic ER-3A) were used to obtain the click thresholds. Signal levels were presented at 70, 50, and 30 dB nHL and wave IV was observed at each presentation level.

### Brain MRI

**Data acquisition.** Neuroimaging data were collected using a 3T MRI scanner (GE750, GE Healthcare, Waukesha, WI) with a Tx/Rx 8-channel volume coil. Data were acquired in a single session with the parameters: repetition time (TR) = 9272 ms, echo time (TE) = 4064 ms, θ = 12˚, field of view (FOV) = 100 × 100 mm, slice thickness = 0.8 mm. Animals were scanned in the supine position in the same orientation, achieved by placement and immobilization of

the head in a custom-made head holder via ear bars. Scans were collected under anesthesia described in S3 Table. End-tidal CO2, inhaled CO2, O2 saturation, heart rate, respiratory rate, blood pressure and body temperature were monitored continuously and maintained during each MRI session.

**Data processing.** For structural MRI data processing and analysis, we used the Auto-Seg_3.3.2 pipeline, developed at the Neuro Image Research and Analysis Laboratories (NIRAL) of the University of North Carolina at Chapel Hill and publicly available on the NITRC website, at http://www.nitrc.org/projects/autoseg. This software pipeline employs BatchMake pipeline scripts that call tools within the AutoSeg toolset, based on the Insight Tool Kit (ITK). T1w DICOMs were initially imported in 3D Slicer (http://www.slicer.org) and saved to NRRD format, and orientation and obliquity of each dataset checked and corrected to LPI (neurological) orientation using the ANTS software [37]. Image processing in AutoSeg proceeded using the step as described in S2 Fig. The first step was to perform an intensity inhomogeneity correction using the N4 algorithm [38]. The second step was to perform rigid-body registration of the subject MRI to the 2-week old UNC-Emory infant rhesus macaque atlas [39] using the 3D BRAINSFit [40] tool within 3D Slicer. The third step was tissue segmentation and skull stripping separating the brain tissues [gray matter (GM), white matter (WM) and cerebrospinal fluid (CSF)] from non-brain image. AutoSeg uses the Atlas Based Classification (ABC) tool [41–43] to perform tissue segmentation as well as skull-stripping integrated into a single method. The fourth step was registration of the atlas to the subject's brain to generate cortical parcellations (affine followed by deformable ANTS registration) [37] to register each skull-stripped atlas image to the skull-stripped subject image using a cross-correlation similarity metric and a symmetric diffeomorphic deformation model that preserves anatomical topology even with large deformation (S2B and S2C Fig). Previous studies have shown that the cross-correlation metric offers enhanced reliability and accuracy as the image registration metric within ANTS [44]. This analysis outputs volumes of brain WM and GM, CSF, and cortical (temporal, prefrontal, frontal, parietal, occipital lobes, and cerebellum) and subcortical (hippocampus, amygdala, caudate, and putamen) regions, which has been previously described [39, 44].

## Infant or fetus necropsy

Infants were sedated and euthanized on DOL 7 or 8. Fetus 416597 was stillborn prematurely and was submitted to necropsy immediately after delivery. Sterile instruments were used for the dissection and collection of all tissues during gross post-mortem examinations. Each tissue was collected with a unique set of sterile instruments and placed in a separate sterile petri dish before transfer to RNAlater for ZIKV qRT-PCR or 4% paraformaldehyde for histology. Ten slices of the fetal/infant cerebrum (~5 mm in thickness) were prepared in the coronal plane with section 1 located most anteriorly and section 10 located most posteriorly, and three slices of the cerebellum were prepared in the sagittal plane with section 1 located most medially and section 3 most laterally; alternate sections were taken for qRT-PCR to analyze vRNA and for histology.

## Histology

Tissues were fixed in 4% PFA other than cerebrum, cerebellum and one eye for histology which were fixed in 10% neutral buffered formalin. Tissues were sectioned (~5 mm), routinely processed and embedded in paraffin. Paraffin sections (5 μm and 8 μm for the brain) were stained with HE or Gram stain using standard methods. The pathologists were blinded to vRNA findings when evaluating and describing tissue sections and assigning morphologic

diagnoses. Photomicrographs were obtained using brightfield microscopes Olympus BX43 and Olympus BX46 (Olympus Inc., Center Valley, PA) with attached Olympus DP72 digital camera (Olympus Inc.) and Spot Flex 152 64 Mp camera (Spot Imaging, Sterling Heights, MI), and captured using commercially available image-analysis software (cellSens DimensionR, Olympus Inc. and Spot 5.3 Software). The ear diagram used to describe where ear sections were obtained from is a composite of multiple histologic sections of infant 020501 with the addition of an external pinna based on gross images.

## In situ hybridization

In situ hybridization (ISH) was conducted either in brain tissues post-fixed in neutral buffered formalin or in non-brain tissues post-fixed in 4% PFA for 24 hours, alcohol processed and paraffin embedded. ISH probes against ZIKV genome were purchased commercially (Advanced Cell Diagnostics, Cat No. 468361, Newark, California, USA). ISH was performed using the RNAscope® Red 2.5 Kit (Advanced Cell Diagnostics, Cat No. 322350) according to the manufacturer's instructions. Briefly, after deparaffinization with xylene, a series of ethanol washes, and peroxidase blocking, sections were heated in antigen retrieval buffer and then digested by proteinase. Sections were exposed to ISH target probe and incubated at 40˚C in a hybridization oven for 2 h. After rinsing, ISH signal was amplified using company-provided Pre-amplifier followed by the Amplifier containing labelled probe binding sites and developed with a Fast Red chromogenic substrate for 10 min at room temperature. Sections were then stained with hematoxylin, air-dried, and mounted. Positive control probes for endogenous rhesus macaque mRNA (Advanced Cell Diagnostics, Cat No. 457711, Newark, California, USA) and negative control probes for bacterial mRNA (Advanced Cell Diagnostics, Cat No. 310043, Newark, California, USA) were used as process controls to verify ISH labelling procedure was successful. Mouse brain tissue served as a positive control for ZIKV ISH in brain tissues and mouse spleen served as a positive control for ZIKV ISH in non-brain tissues.

## Statistical analysis

Standardized differences, i.e., effect size d's [45], were calculated for quantifying the differences between ZIKV-exposed and control neonates in specific quantitative infant exam parameters. Due to the small sample size in the control group, only the standard deviations of the ZIKV-exposed group observations were used for the effect size estimation. Effect sizes d of 0.2 were considered as small, 0.5 as moderate and >0.8 as large (45). Due to the small sample sizes of the current study, detecting statistically significant differences in the quantitative infant exam parameters was not anticipated. However, the current study provided data to evaluate the distribution of effect sizes for comparing differences ZIKV-exposed and control neonates in quantitative infant exam parameters. We conducted in depth post-hoc power and sample size calculations to define the sample sizes required to discern differences between ZIKV-exposed and control neonates in quantitative infant exam parameters which might be useful for the planning of future studies. These specific quantitative infant parameters included: scores for each time point and construct in the neonatal neurobehavior assay (SNAP), each ocular layer thickness (OCT), each wave characteristic in the visual electrophysiology assays (ERG and VEP), infant weight gain trajectory, infant cumulative feeding volume, and each brain region volume (structural MRI). Sample size estimates were performed to detect observed effect sizes with 80% power at the two-sided 0.05 significance level and were based on a two-sample t-test with equal variances and assuming either 1:1 or 2:1 sample size allocations between the ZIKV-exposed and control groups. Both sample size allocations were selected (1:1 and 2:1) because some study designs have more ZIKV-exposed infants than control infants. The post-hoc

power analysis was conducted using SAS (SAS Institute, Cary, NC, version 9.4) (S1 Appendix) and histograms were generated using R statistical language (R Development Core Team 2019), version 3.6.3.

## Study approval

The University of Wisconsin-Madison, College of Letters and Science and Vice Chancellor for Research and Graduate Education Centers Institutional Animal Care and Use Committee approved the nonhuman primate research covered under protocol number G005401-R01.

# Results

## ZIKV inoculation during pregnancy

Five dams were inoculated with ZIKV and one dam was inoculated with PBS in the first trimester (41–50 gestational days (gd)), as described in Table 1 above. No dams had fever, rash, or inappetence following inoculation. One dam had a stillbirth at gestational day 133 and the fetus was found partially delivered. The remaining infants were delivered by cesarean delivery at 154–155 gd (term = 166.5 gd) (45) (Fig 2) to ensure collection of maternal-fetal interface (MFI) tissues. Maternal plasma viremia peaked 3–5 days post inoculation, and 4 of 5 animals had extended plasma viremia for at least 28 days (S3A Fig). The dams also developed the expected immune response to the virus: ZIKV-specific neutralizing and binding antibodies were detected at 28 dpi and persisted until the last time point measured at necropsy (S3B and S3C Fig, S4 Fig).

## Pregnancy course and fetal ultrasonography

None of the fetuses had microcephaly (defined as less than 2 SD below the mean) in the ultrasound immediately prior to delivery, even though there was variability in head circumference compared with the normative mean throughout gestation (Fig 3). The fetuses also did not have a femur length or abdominal circumference that was greater than 2 SD below the mean just prior to delivery (S5A and S5B Fig). The single fetal demise (664184) was identified by an

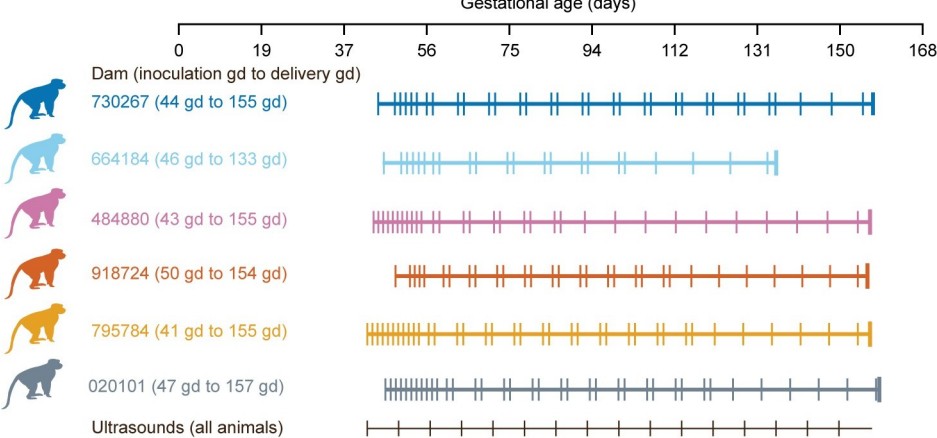

**Fig 2. Pregnancy experimental design, maternal plasma vRNA loads and maternal antibody responses.** Pregnant macaques were inoculated with ZIKV-PR (all colors) or PBS (grey color) in the first trimester. One female had a stillbirth at 133 gestational days (gd). Maternal blood was obtained at the times indicated for each animal (vertical lines on each animal's timeline) and ultrasound analyses of the fetus and placenta were performed weekly (black vertical lines).

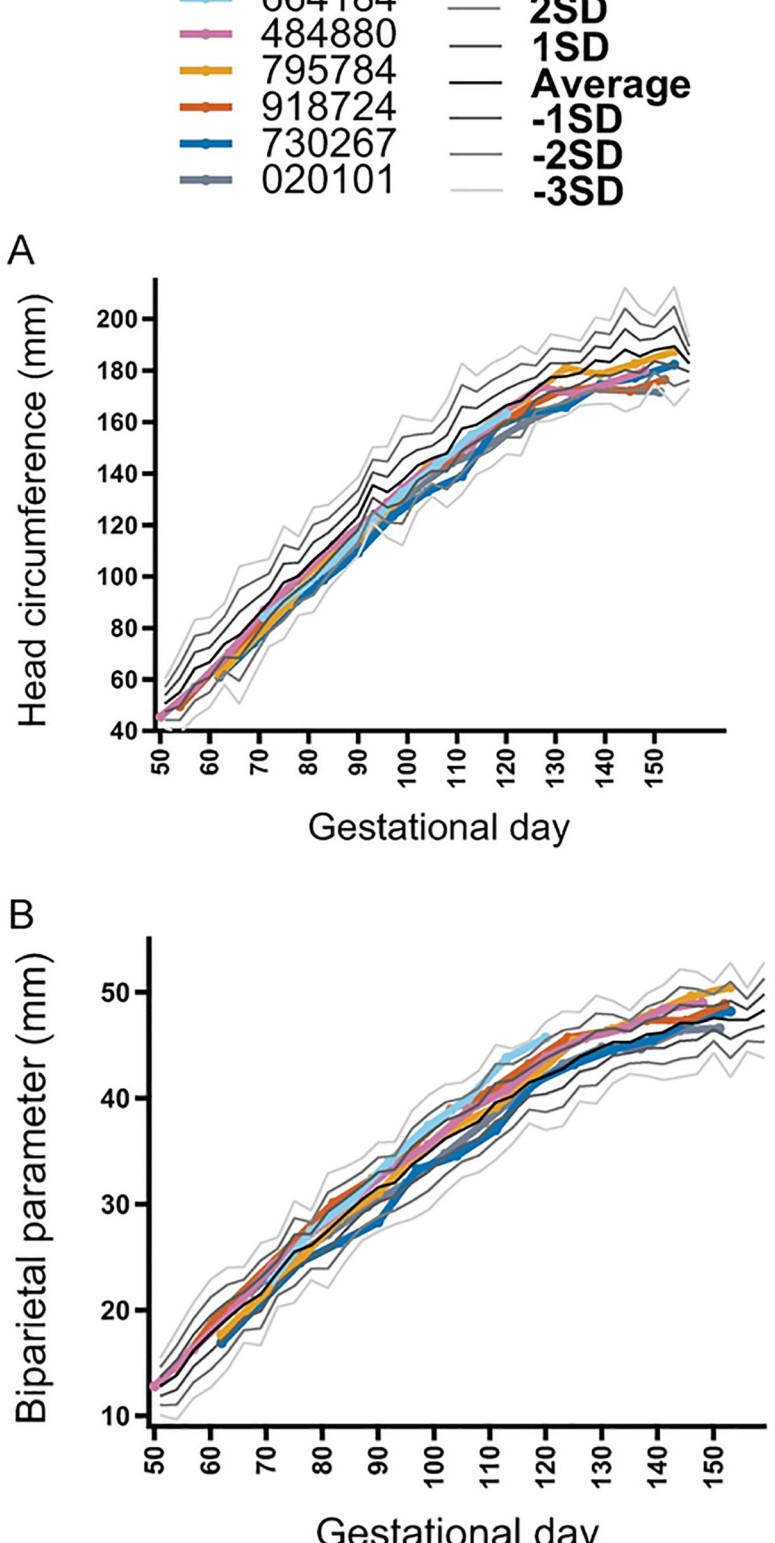

**Fig 3. Fetal head growth during pregnancy.** Fetal ultrasonography depicting head circumference (A) and biparietal diameter (B) during gestation. The growth curve standard deviation (SD) was calculated from normative fetal rhesus macaque measurements [22].

absent fetal heart rate, and this was preceded by decreasing fetal heart rate (bradycardia) from gd85 until stillbirth at gd133 when the fetus (416597) was found partially delivered (S5C Fig). The remaining four fetuses had no life-threatening events in utero.

## Maternal-fetal interface vRNA tissue distribution and histopathology

Histologic evaluation of maternal-fetal interface (MFI) tissues revealed minimal to moderate inflammation in the ZIKV-infected pregnancies (Fig 4 and S4 Table), whereas the control pregnancy had none. The ZIKV-inoculated and control dams revealed some degree of maternal vascular malperfusion, a common finding in the macaque placenta [46, 47]. One dam, 664184, had a large uterine diverticulum (outpouching of the uterine wall) with severe segmental uterine infarction, necrosuppurative inflammation, and ischemic necrosis, confounding interpretation of the stillbirth.

Two of the five ZIKV-exposed animals had detectable RNA (by qRT-PCR or ISH) in the maternal-fetal interface tissues at the time of delivery (Fig 5). The decidua of ZIKV-exposed dam 795784 had ZIKV RNA of 56 copies/mg detected by qRT-PCR. ZIKV-exposed dam 644184 had ZIKV RNA detected by ISH in the decidua and amniotic/chorionic membrane. However, ZIKV RNA was not detected in the uterine placental bed, uterus, vagina, placental discs, umbilical cord or amniotic fluid.

## Infant clinical courses

Four of the five ZIKV-exposed and control infants required noninvasive respiratory support during resuscitation efforts after delivery (S5 Table); it is unclear whether the need for respiratory support at this premature gestational age is unusual because there is not a large cohort of infants delivered before gd165 by cesarean section at WNPRC available for comparison. One infant had significant respiratory disease (499874), requiring noninvasive respiratory support for almost a day after delivery, and then again following sedated exams. No clinical etiology of for the respiratory symptoms were identified as the chest x-ray was unremarkable. Following placement in the nursery, all infants underwent a rigorous schedule of qualitative and quantitative assessments as described in Fig 1.

## Infant viral loads and antibody responses

We did not identify a ZIKV-specific IgM response in the ZIKV-exposed infants (S6A and S6B Fig). Plasma, urine and CSF viral loads were also negative in the ZIKV-exposed infants (S6C Fig). Neutralizing antibody concentrations in infants are comprised of vertically transmitted maternal IgG antibodies and some fetal-derived IgG [48]. The neutralizing antibody concentration in the infants was lower than the maternal neutralizing antibody concentration in all the ZIKV-exposed infants (S6D Fig, with infant neutralizing antibody dilution curves in S7 Fig).

## Infant growth and feeding volume sample size estimation

Infant weight gain and formula consumption were monitored closely to evaluate for secondary effects of dysphagia, or difficulty swallowing, which affects some children with congenital Zika syndrome [49]. We demonstrate that all the infants gained weight and had increasing cumulative feeding volumes during their first week of life (Fig 6A and 6B). In order to determine whether our study was powered to detect differences in weight gain and feeding volume trajectories between groups, we conducted post-hoc power and sample size calculations based on observed effect sizes. Large effects sizes were observed for both weight gain trajectory

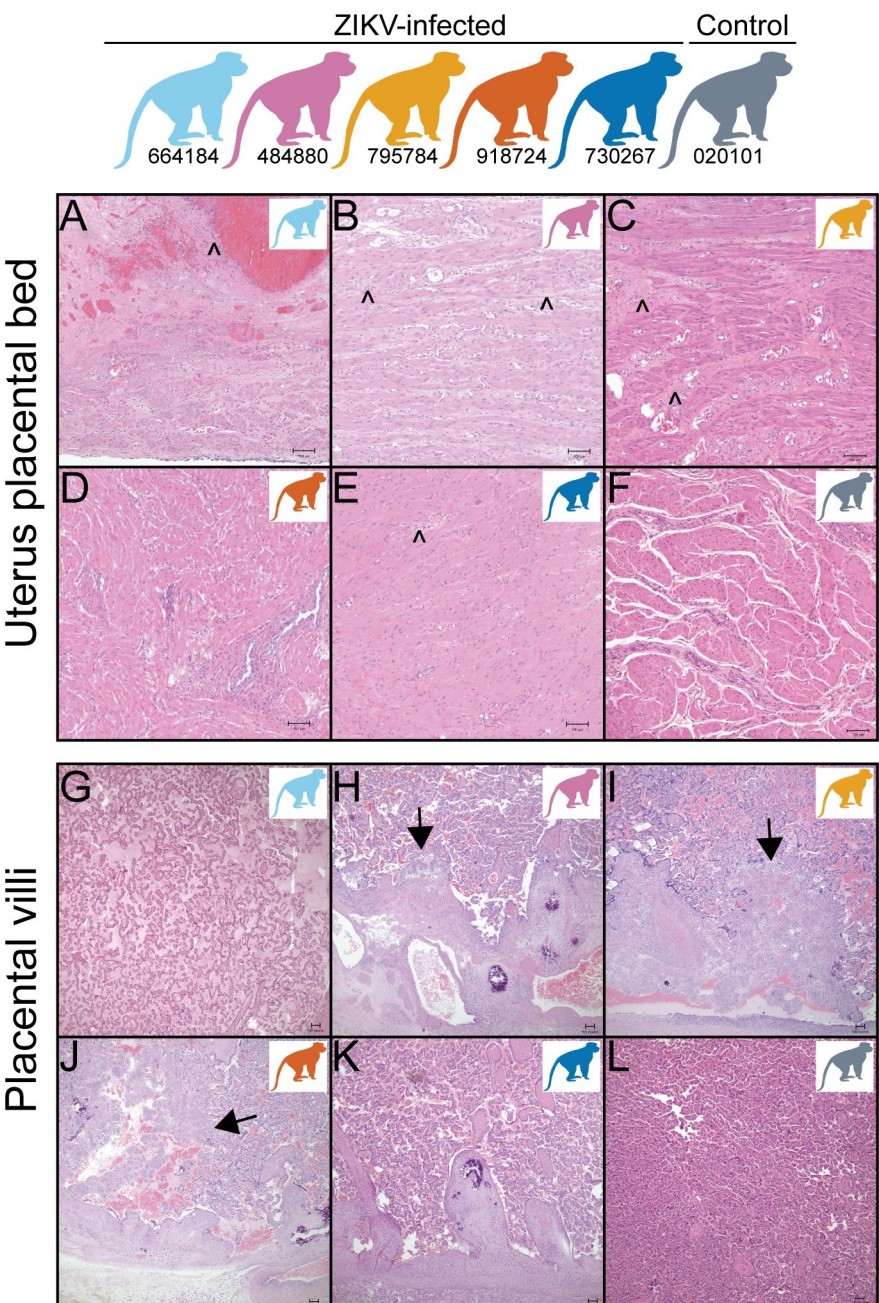

**Fig 4. Placental bed and placental villous pathology.** (A-F) Uterine placental bed histology reveals minimal to severe lymphoplasmacytic myometritis or neutrophilic to lymphoplasmacytic endometritis (caret symbol) in 4 of 5 ZIKV-exposed dams (A, B, C, E) but not in one of the ZIKV-exposed dams (D) or control decidua (F). (G-L) Within the placenta, 3 of the 5 ZIKV-exposed pregnancies have villitis (arrow heads) (H, I, J) but not in 2 of the ZIKV-exposed dams (G, K) or a control placental villi (L). Colors of macaques in each image represent individual animals as depicted at the top of the figure. Scale bar is 100 μm.

(d = 1.23) and cumulative feeding volumes (d = 0.9). We demonstrate that the sample size required to define differences between the weight gain trajectories in ZIKV-exposed and control infants is 12 infants per group (Fig 6C). A larger group of infants (n = 21 per group) is required to define significant differences between cumulative feeding volumes (Fig 6C).

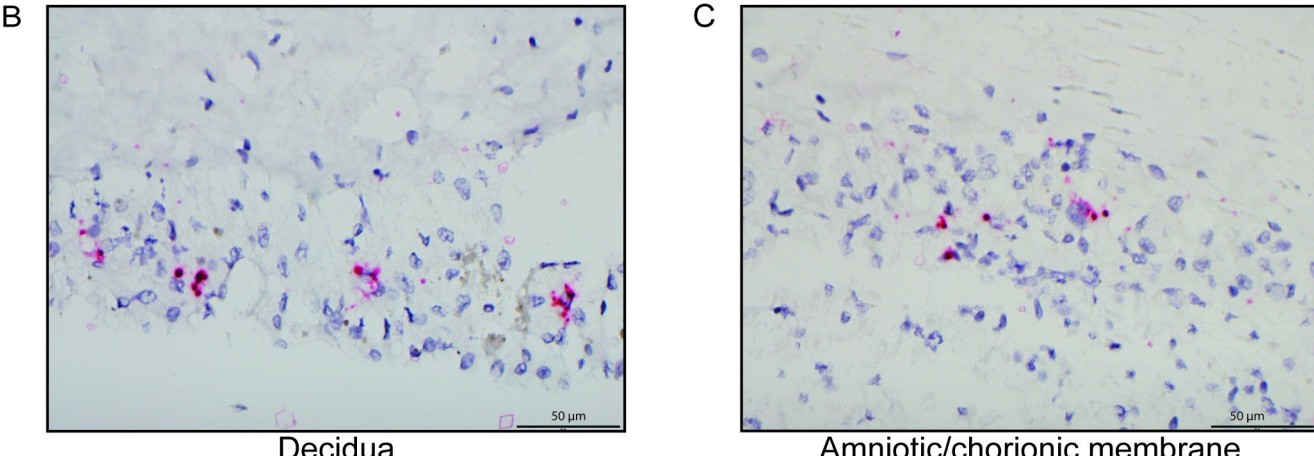

| Tissue Source | Tissue Name | Animal ID (dam ID/fetus-infant ID) | | | | | | | | | |
|---|---|---|---|---|---|---|---|---|---|---|---|
| | | 664184/416597 | | 484880/424847 | | 795784/499874 | | 918724/527421 | | 730267/226691 | |
| | | qRT-PCR | ISH | qRT-PCR | ISH | qRT-PCR | ISH | qRT-PCR | ISH | qRT-PCR | ISH |
| Maternal tissues | Decidua | ND | positive | ND | negative | 56 copies vRNA/mg | negative | ND | negative | ND | negative |
| | Placental bed | ND | NT | ND | NT | ND | NT | ND | NT | ND | NT |
| | Uterus | ND | NT | ND | NT | ND | NT | ND | NT | ND | NT |
| | Vagina | ND | NT | ND | NT | ND | NT | ND | NT | ND | NT |
| Fetal extraembryonic tissues | Amniotic/chorionic membrane | ND | positive | ND | negative | ND | negative | ND | negative | ND | negative |
| | Placental disc 1 | ND | negative | ND | negative | ND | negative | ND | negative | ND | negative |
| | Placental disc 2 | ND | negative | ND | negative | ND | negative | ND | negative | ND | negative |
| | Umbilical cord | ND | NT | ND | NT | ND | NT | ND | NT | ND | NT |
| | Amniotic fluid | NC | NA | ND | NA | NC | NA | ND | NA | ND | NA |

Decidua

Amniotic/chorionic membrane

**Fig 5. Viral loads and ISH in the maternal-fetal interface.** (A) Viral loads were assessed in female reproductive and fetal extraembryonic tissues by qRT-PCR at delivery and sections of some of these tissues were assessed by ISH. (B) Positive ZIKV ISH in the decidua of dam 664184. (C) Positive ISH staining in the amniotic/chorionic membrane of fetus 416597. ND = not detected (below the limit of detection), NT = not tested, NC = not collected, NA = not applicable.

## Infant neurobehavioral development

The ZIKV-exposed infants had similar trending scores as the control infant in multiple domains of neonatal neurobehavior, including orientation, motor maturity and activity, sensory and state control (Fig 7). The only domain in which the ZIKV-exposed infants had a lower trending score than the control infant was in one of the time points of the orientation construct (Fig 7A), which assesses both visual and auditory orientation skills. We did not statistically define differences between ZIKV-exposed and control groups because of our small sample size and because published normative data for neonatal neurobehavior do not match our infant population closely enough to be useful as a comparator due to the early delivery in our sample [29, 30]. To determine what sample size will be necessary in future studies to statistically define differences between ZIKV-exposed and control groups, we calculated effect sizes and performed sample size assessments for each construct and time point, resulting in 12 effect and sample size calculations (S6 Table). We developed a histogram showing the distribution of the effect sizes (Fig 7E). The histogram demonstrates that 25% of the effect sizes are >1.5 (Fig 7E), which corresponds to a sample size of 8 infants or less per group required to statistically define differences between ZIKV-exposed and control infants (S6 Table). The constructs with the most difference between groups, i.e. effect size > 1.5, included time points in the motor

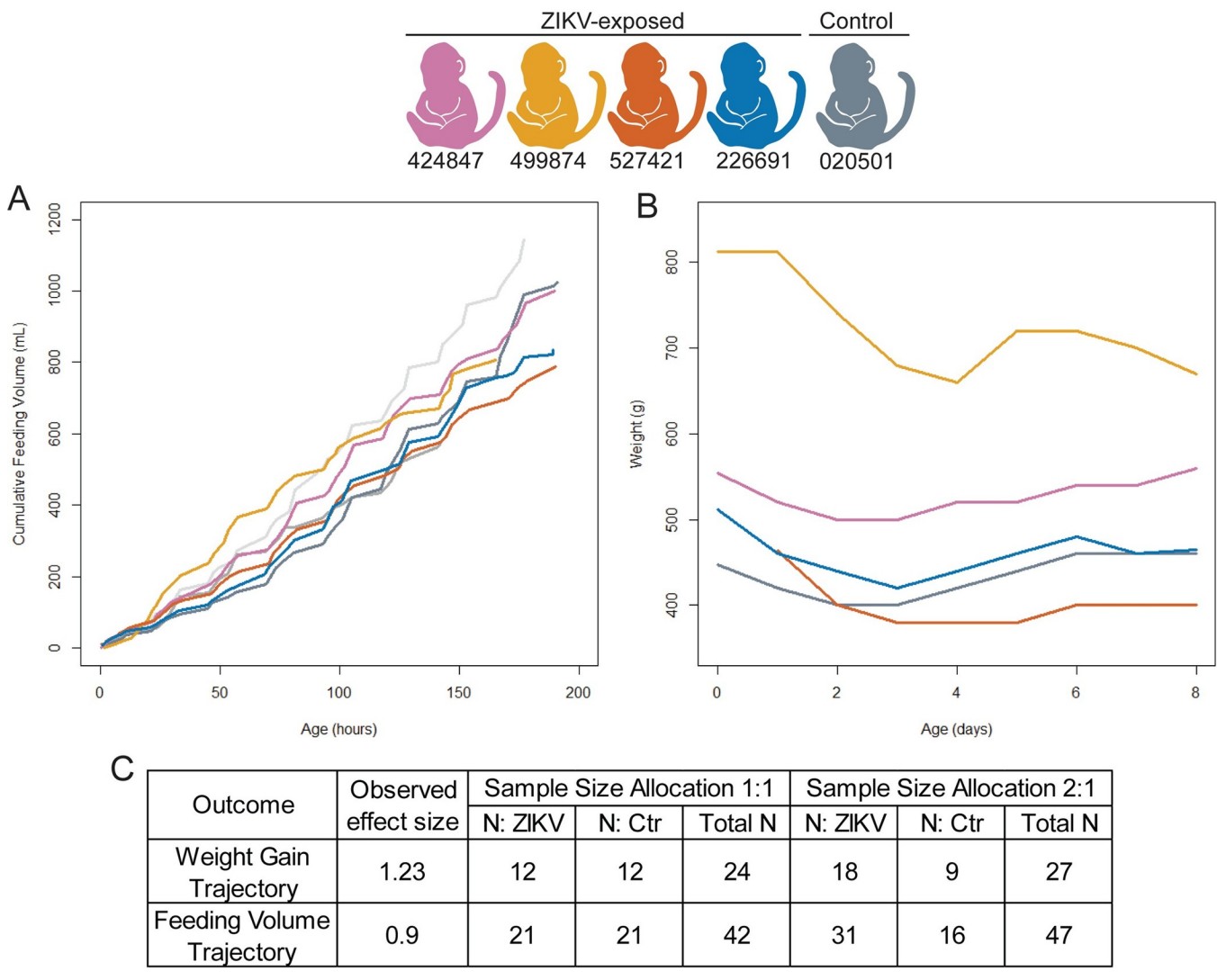

**Fig 6. Infant weight gain, cumulative feeding volumes, and sample size estimates.** Cumulative feeding volumes (A) and weights (B) were measured during their first week of life. Observed effect sizes and sample size requirements (C) for detecting observed effect sizes for weight gain and feeding volume trajectory with 80% power at the two-sided 0.05 significance level, assuming a sample size allocation of 1:1 and 1:2 ZIKV to control animals.

maturity and activity construct, and the orientation construct (S6 Table). This means that sample sizes of at least 8 animals per group are needed to statistically define differences in 25% of neonatal neurobehavioral parameters.

## Infant vision evaluation

Two of the ZIKV-exposed infants had minor ocular defects similar to defects observed in human infants with congenital ZIKV infection identified with indirect ophthalmoscopy, a corneal defect and retinal pigmented epithelium mottling (S7 Table). No other ocular birth defects associated with congenital Zika syndrome were identified [50], including no optic nerve hypoplasia, choroidal lesions, lens abnormalities or vitreous opacities. We next quantified the thickness of each ocular layer in the anterior and posterior segments of the eye by ocular coherence tomography (Fig 8A) because subtle ocular defects that impact vision may not be apparent on ophthalmic exam. Full thickness ocular layers (choroid, cornea and retina) did

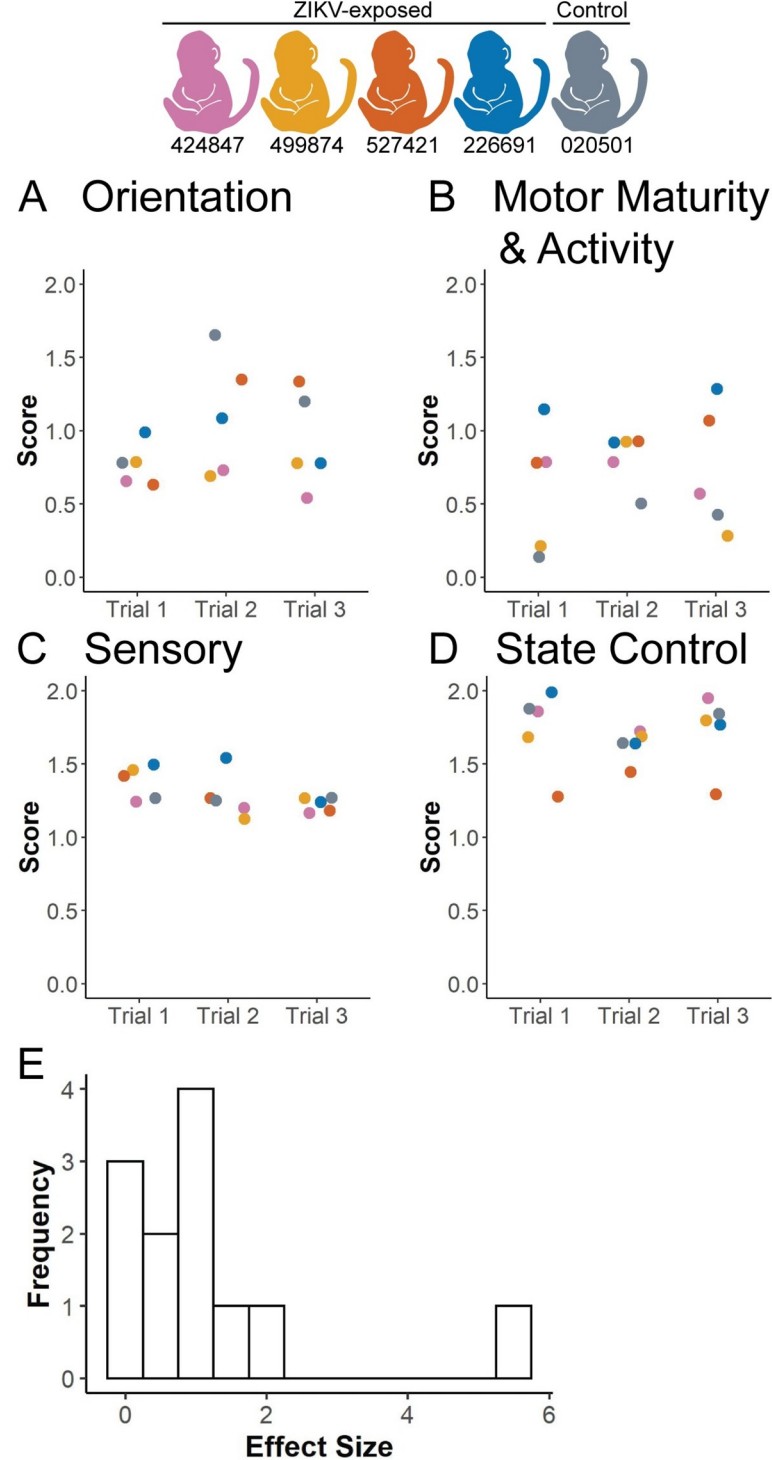

**Fig 7. Neurobehavioral assessment.** The (A) Orientation construct, (B) Motor Maturity and Activity construct, (C) Sensory construct and (D) State Control construct were assessed on three separate days (Trials 1–3) during the infants' 8 days of life before necropsy, as described in S2 Table. Test items are scored from 0–2, with 0 representing no response or performance and 2 representing a higher performance of the measured behaviors. (E) A histogram illustrates the distribution of effect sizes for all four constructs at each of the three time points.

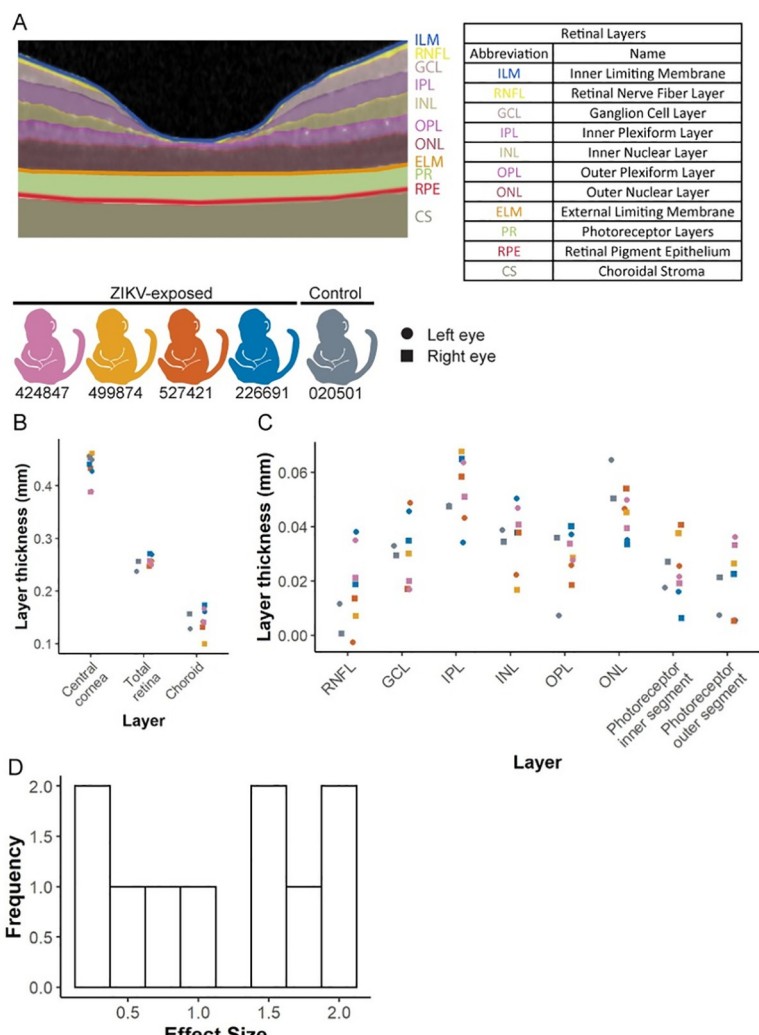

**Fig 8. Ocular and retinal layer thicknesses measured by optical coherence tomography.** (A) Retinal layer segmentation was performed using EXCELSIOR Preclinical, and layers are marked as an example from infant 424847. (B) Thicknesses of the retina, central cornea and choroid were measured in the ZIKV-exposed and control infants with the designated color scheme. Values from the left and right eyes are demonstrated with a circle and a square, respectively. (C) Retinal layer thicknesses were measured using the segmentation demonstrated in part A for ZIKV exposed and control infants. Both the right and left eyes were measured in all animals except for animal 499874 whose left eye was not able to be measured due to anesthesia limitations. (D) A histogram illustrates the distribution of effect sizes for all ocular layer thicknesses.

not display much variability between ZIKV-exposed and control animals (Fig 8B). In contrast, there is more variation in the thickness of individual retinal layers between animals (Fig 8C). We did not define differences between groups because of our small sample size, and instead estimated effect and sample sizes needed to define differences for future studies. An effect size of >1.5 corresponds to a sample size of 8 infants or less per group required to statistically define differences between ZIKV-exposed and control infants (S8 Table). Of the 10 ocular layer thicknesses we measured in the anterior and posterior segment, 4 of the 10 layers had an effect size of >1.5 as shown in a histogram in Fig 8D. The layers with the largest effect sizes are the ganglion cell layer (GCL), inner plexiform layer (IPL), outer nuclear layer (ONL) and photoreceptor outer segment layer (S8 Table). This means that sample sizes of at least 8 animals

per group are needed to statistically define differences in 4 of the 10 ocular layer parameters. When considering all 10 ocular layer thicknesses together, the total number of animals needed to define differences in 50% of the parameters is 13 animals per group, with an observed effect size of 1.16.

Visual function may be altered in structurally normal eyes, so we assessed retinal function using electroretinography and visual evoked potentials to assess the cortical visual pathway (Fig 1). The ZIKV-exposed and control infants demonstrated no apparent trends of the retinal function or cortical visual function in electroretinography (Fig 9A) or visual evoked potential (Fig 9B) studies, respectively. As with the other quantitative neonatal assessments, we defined effect and sample sizes needed to statistically define differences in future studies. An effect size of >1.5 corresponds to sample size estimates of 8 animals needed per group to statistically define differences between ZIKV-exposed and control infants with 80% power at the two-sided 0.05 significance level (S9 Table). Two of the 18 visual pathway parameters assessed had effect sizes of >1.5 (Fig 9C). The visual pathway parameters with an effect size >1.5 are the A wave amplitude and latency. When considering all 18 electroretinography and visual evoked potential parameters together, the total number of animals needed to define differences in 50% of the parameters is 22 animals per group, with an observed effect size of 0.87.

## Infant hearing evaluation

We assessed hearing quantitatively with auditory brainstem response testing (Fig 1) because hearing loss is one of the findings in congenital Zika syndrome [51]. One ZIKV-exposed and one control infant demonstrate a similar trend of auditory brainstem evoked potential generated by click stimuli, as demonstrated by the wave IV latency (Fig 10A). These one week old infants display wave IV latencies similar to other neonatal macaques [52], suggesting that they had normal hearing. Only two infants underwent auditory brainstem response testing because of equipment availability and group differences were not statistically defined. Instead, we defined the number of animals that will be needed in future studies to statistically define differences between ZIKV-exposed and control groups in each of the 6 parameters (3 click intensities on the left and right sides). All of the hearing parameters had an effect size of 1.41, given the small number of infants, which corresponds to a sample size estimate of 9 animals needed per group to statistically define differences between ZIKV-exposed and control infants with the observed effect sizes.

## Infant brain imaging

We found no severe brain abnormalities consistent with congenital Zika syndrome [53] in our ZIKV-exposed or control infants, including subcortical calcifications, ventriculomegaly, cortical thinning, gyral pattern anomalies, cerebellar hypoplasia or corpus callosum anomalies on qualitative neuroradiological interpretation. Next, we evaluated for subtle changes that would not be apparent on qualitative neuroradiological interpretation by defining structural brain region volumes of the cortical and subcortical regions (Fig 2B and 2C). A total of 40 brain region volumes were quantified, including the total volume of regions (cortical, subcortical, CSF) relative to total brain volume (TBV) or intracranial volume (ICV) to take into account the infant's brain and head size, respectively (S10 Table). The single control infant had the largest total brain, CSF and lateral ventricle volume of all the infants (Fig 11A), even though it was the youngest infant at the time of the MRI (020501 was 159 corrected gestational days, whereas the ZIKV-exposed infants were 161–163 corrected gestational days old) (S1 Table). The control infant also had the largest relative total white matter volume as corrected for total brain volume, and the smallest grey matter volume (Fig 11B). Volumetric brain analyses at this

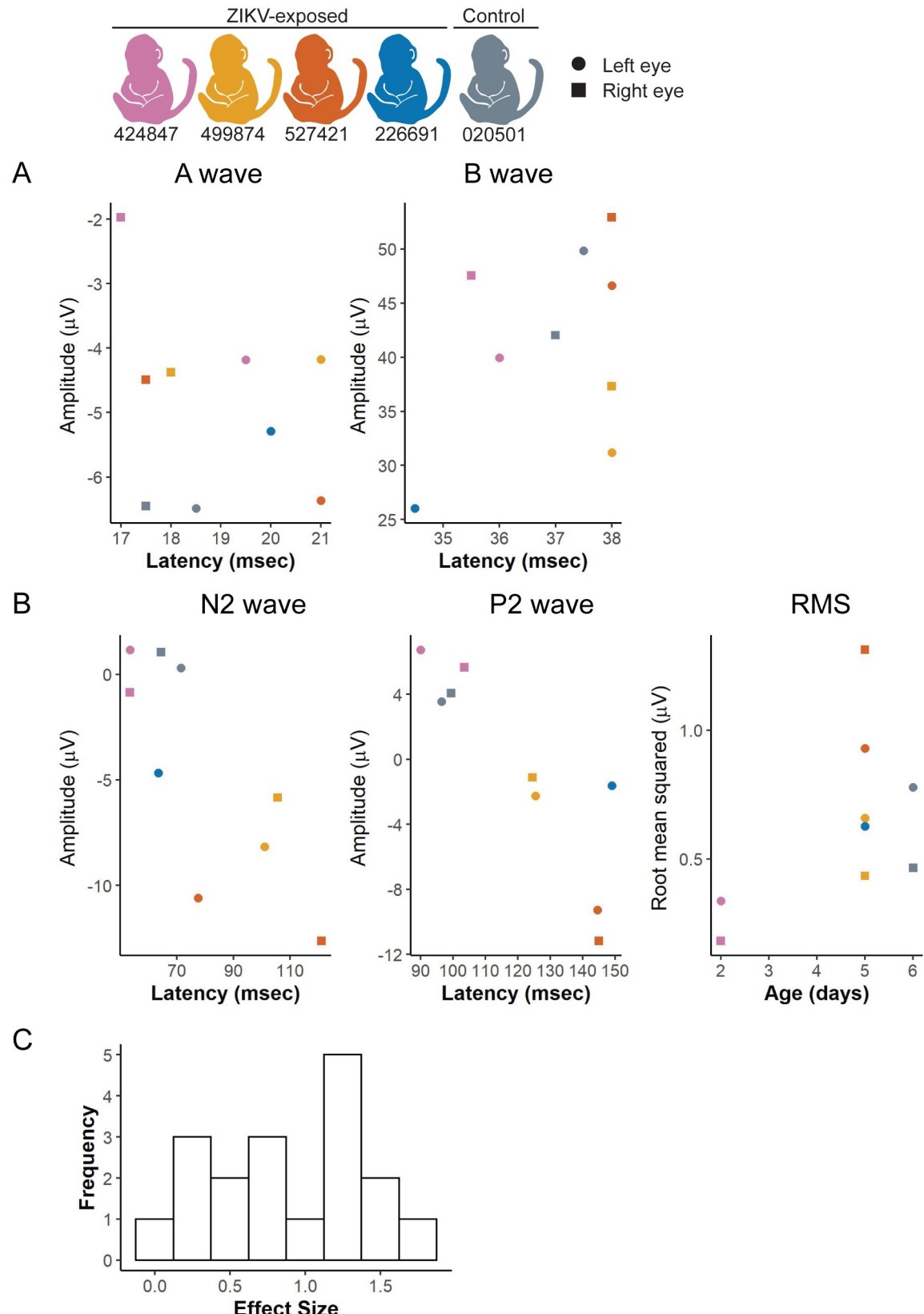

**Fig 9. Visual function measured by electroretinography and visual evoked potentials.** (A) Light-adapted ERGs were performed for each infant and the amplitude and latency of the A and B waves were recorded. Right and left eye values were graphed separately with a square or circle, respectively. (B) Visual evoked potentials (VEP) were performed on an uncovered right or left eye, and the amplitude, latency and root-mean-square (RMS) of the N2 and P2 waves were recorded. (D) A histogram illustrates the distribution of effect sizes for each ERG, VEP, and RMS value.

young neonatal age are challenging because the grey to white matter contrast is limited, and future studies could improve the imaging results by acquiring and averaging multiple runs of T1w and T2w images. The biological significance of these volume differences cannot be determined in this pilot study; longer term studies are being pursued to evaluate these findings.

To define how many infants will be required in future studies to detect differences between ZIKV-exposed and control groups, we calculated effect sizes and conducted post-hoc sample size calculations. Ten of 40 brain region volumes quantified (total volumes and TBV corrected) had effect sizes of >1.5 (Fig 11C), which corresponds to a sample size of 8 infants per group required to sufficiently define differences between ZIKV-exposed and control infants (S10 Table). The brain regions with the largest effect sizes included the total brain volume and relative white matter, gray matter, right temporal auditory region and right temporal visual region corrected by total brain volume (S10 Table). The intracranial volume correction shows similar results as the total brain volume correction, with 10 of 40 brain regions demonstrating effect sizes of >1.0 (S10 Table). In summary, this feasibility study demonstrates that sample sizes of at least 8 infants per group are sufficient to define differences in 25% of the brain region volumes.

## Infant tissue vRNA distribution and histopathology

No ZIKV vRNA was identified by qRT-PCR in over 45 tissues and body fluids examined per infant/fetus at necropsy (S11 Table), including multiple brain sections. Brain sections were also examined by in situ hybridization (ISH) for ZIKV RNA, and there was no positive staining in any of the fetus/infant brain sections evaluated (S8 Fig). We also searched for ZIKV RNA by ISH of the infant/fetus lung tissue, as this tissue had high viral loads in our previous studies [15], and in the cochlea, because of the clinical findings of sensorineural hearing loss in human congenital ZIKV infection, however no fetal/infant lung or cochlea section had positive ZIKV ISH. The control and ZIKV-exposed infants had similar morphometric measurements at necropsy (S12 Table).

Histopathology demonstrated two key findings in the ZIKV-exposed infants but not control infant: neutrophilic otitis media (i.e. middle ear inflammation) in 3 of 5 ZIKV-exposed fetuses/infants (Fig 12) and bronchopneumonia in 2 of 5 ZIKV-exposed infants (Fig 13), summarized in S13 Table. The infant who had respiratory distress after birth and with each sedation event (499874) had an unremarkable chest x-ray, but bronchopneumonia was identified histologically. The other infant with histological bronchopneumonia (226691) had no respiratory symptoms. Together, this means that 4 of the 5 ZIKV-exposed infants either had neutrophilic otitis media or bronchopneumonia, and only one of these infants displayed signs of illness. There were no lesions identified by hematoxylin and eosin (HE) staining in the eye, brainstem or CNS tissues of any of the ZIKV-exposed infants or in the control infant; additionally, no remaining tissues had lesions that affected more than one fetus/infant (all tissue histopathology described in S14 Table). HE stained brains had no significant histologic lesions; quantitative analyses are in progress.

## Quantitative infant exam sample size estimations

Comprehensively defining infant neurodevelopment and underlying abnormalities requires understanding how many infants will be needed to define differences between ZIKV-exposed

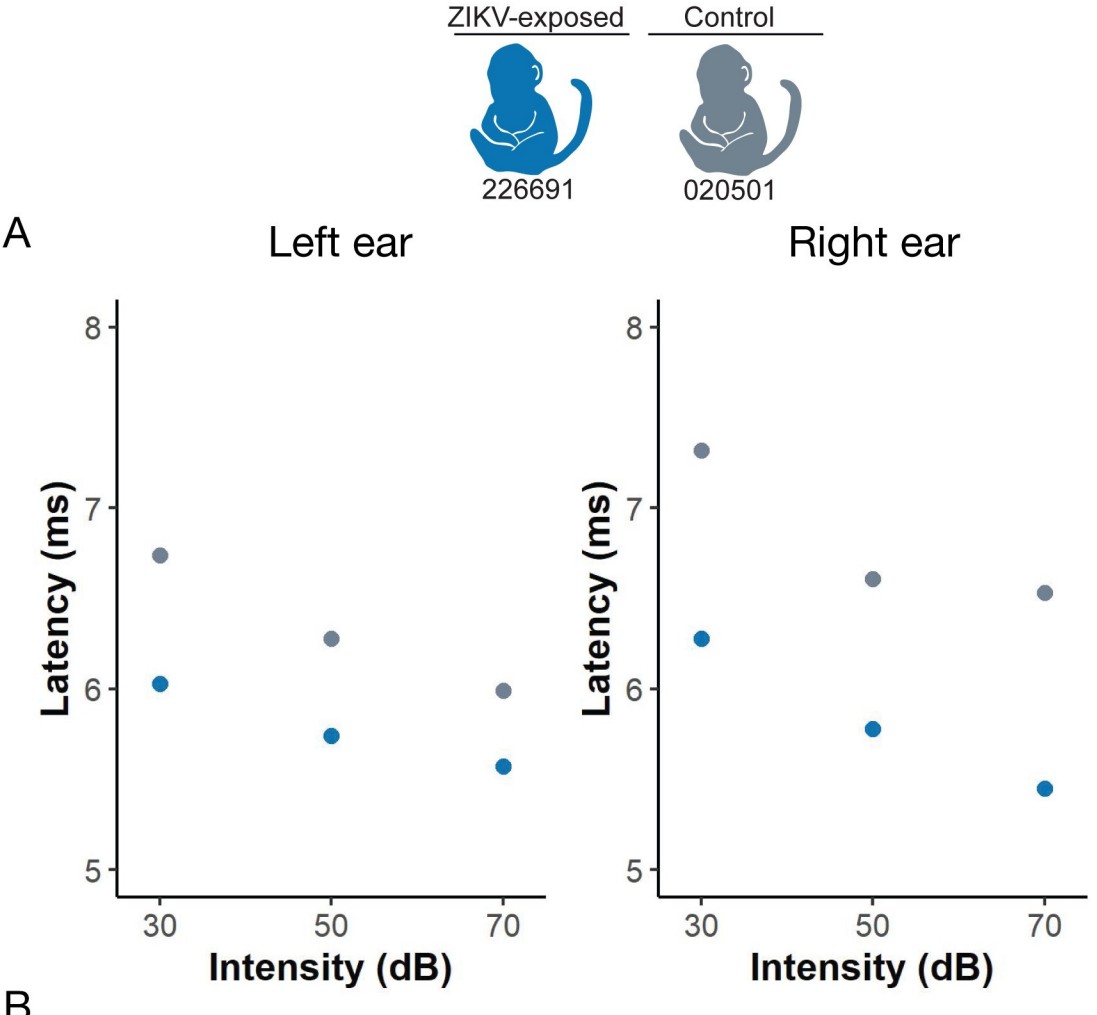

**Fig 10. Hearing measured by auditory brainstem response testing.** (A) Auditory brainstem response testing Wave IV thresholds to click stimuli. (B) Effect sizes and sample size requirements for detecting the observed effect sizes with 80% power at the two-sided 0.05 significance level, assuming a sample size allocation of 1:1 and 1:2 ZIKV to control animals.

and control groups. When all the quantitative infant exams are combined, a total of 76 individual parameters were quantified (S15 Table). A sample size of 8 animals per group is adequate for defining statistically significant differences in 22% of all the quantitative infant exam parameters, assuming similar observed effect sizes to our population. If the sample sizes

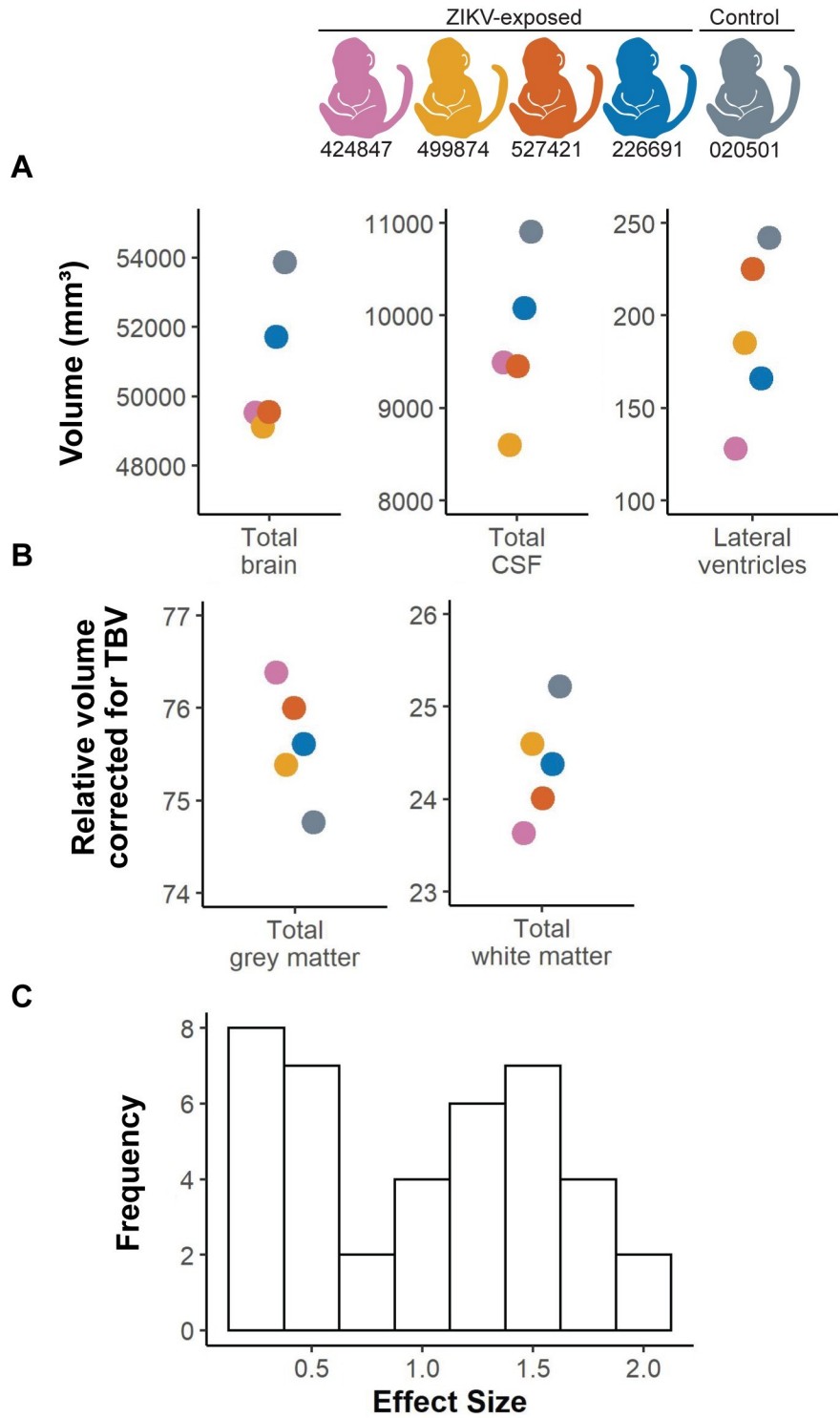

**Fig 11. Volumetric brain analysis and effect sizes.** (A) Structural MRI measurements in cubic millimeters for total brain volume (TBV), lateral ventricles and cerebrospinal fluid (CSF). (B) Grey matter and white matter specific areas corrected for TBV in all infants. (C) A histogram illustrates the distribution of effect for entire brain regions and regions corrected by TBV (tissue type, cortical regions, subcortical regions).

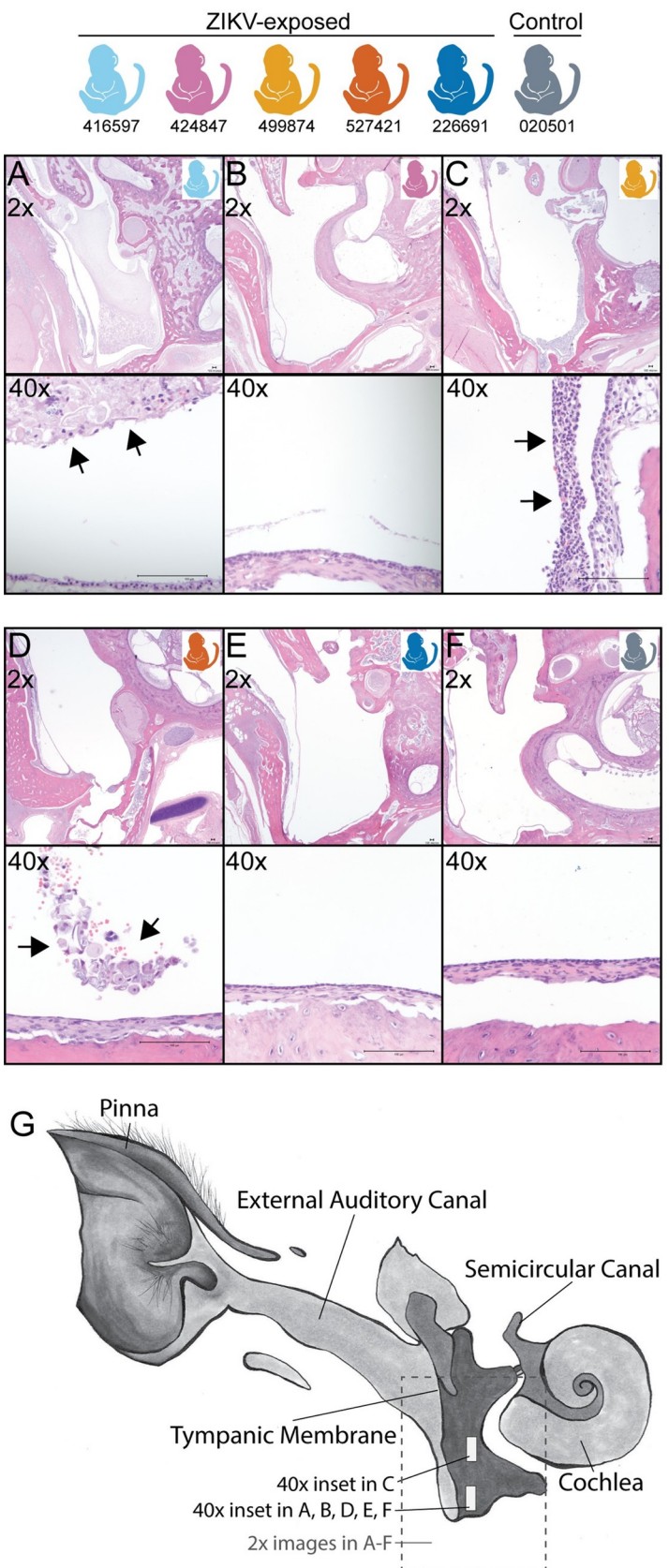

**Fig 12. Fetus/Infant tissue histopathology of the ear.** Neutrophilic otitis media (40x inset) is present in 3 of 5 ZIKV-exposed fetus/infants (A, C, D) and not in a control infant (F) or two ZIKV-exposed infants (B, E). Arrowheads denote the location of the neutrophilic inflammation. (G) Ear diagram depicting where tissue sections were obtained for the 2x and 40x magnifications. Macaque colors represent animals as depicted at the top of the figure. Scale bar is 100 µm.

increases to 14 animals per group, 51% of the exam parameters will have adequate sample sizes to define statistically significant differences.

## Discussion

Our results indicate that it is feasible to define infant development quantitatively with moderately sized studies. These quantitative infant exam results generate testable hypotheses about visual pathway abnormalities in congenitally ZIKV-exposed infants, which can be assessed in future appropriately powered studies. We also demonstrate that ZIKV vRNA is not identified in all ZIKV-exposed infants and identify a novel pathological finding in ZIKV-exposed infants: neutrophilic acute otitis media and bronchopneumonia. This battery of quantitative infant macaque-adapted exams complemented by histopathological analyses may be useful for defining early predictors of abnormal development in congenital ZIKV infection and other congenital infections, such as cytomegalovirus.

This model of congenital ZIKV exposure is similar to human clinical studies where maternal infection is confirmed and no ZIKV vRNA or IgM is detected in the infants [21, 54–56]. The lack of ZIKV vRNA or IgM in infants may occur if fetuses clear the infection before delivery and/or clear an early IgM response before delivery. Another reason may be a lower

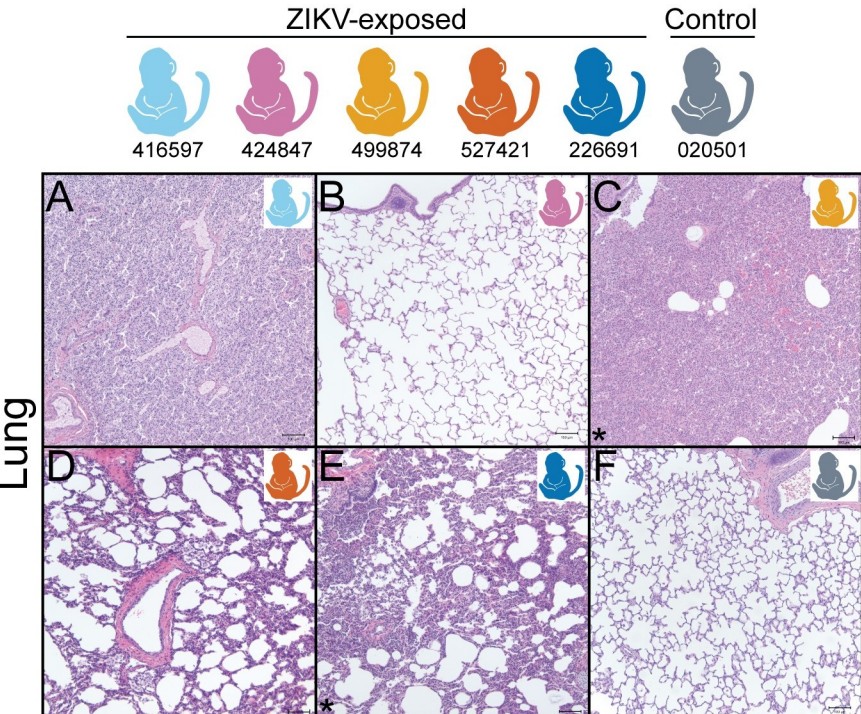

**Fig 13. Fetus/Infant tissue histopathology of the lung.** (A-F) Diffuse bronchopneumonia is present in 2 of 5 ZIKV-exposed infants (C, E) but not in a control infant (F) or 3 of the ZIKV-exposed infants (A, B, D) (images with bronchopneumonia are denoted with an asterisk). The lung in the stillborn macaque (A) is atelectatic (uninflated) due to stillbirth. Macaque colors represent individual animals as depicted at the top of the figure. Scale bar is 100 µm.

inoculation dose and earlier inoculation compared with the only other macaque model that has isolated ZIKV vRNA in liveborn macaque [18]. We identified ZIKV vRNA in the maternal-fetal interface of 2 of the 5 dams at delivery, a similar pattern to other macaque studies [15–17, 57, 58], demonstrating vRNA persistence in some compartments for a long duration. The lack of vRNA and IgM in infant body fluids occurs commonly in human infants exposed to ZIKV in utero [59–61]. This is why human clinical trials do not require evidence of ZIKV RNA in infant body fluids or ZIKV-specific IgM to enroll infants into studies characterizing the long term outcomes of children born to women with documented or suspected ZIKV infection during pregnancy. The challenge in confirming infant infection following documented maternal infection, especially in cases where ZIKV vRNA is present in the maternal-interface at the time of delivery, indicates that better tools to diagnose infant exposure/infection need to be developed.

This is the first time that sample size estimates have been reported for quantitative examinations in infant macaques, although there are limitations to our sample size estimates. A limitation of our sample size calculations is that they are based on measurements from only five infants (4 ZIKV-exposed and 1 control) and were performed only in the early neonatal period. We concentrated our exams in this early period because we wanted to define ZIKV vRNA tissue tropism in neonates before infection was cleared and wanted to associate viral tropism with the development of birth defects. Given that we did not identify any ZIKV vRNA in the infants and did not identify any severe birth defects, we felt it was a better use of precious resources to assign future pregnant animals to studies of long-term neurodevelopment, rather than include additional control animals in this short-term study. Studies of long-term neurodevelopment are better suited for macaque studies because the incidence of congenital ZIKV-associated developmental deficits is more common than congenital ZIKV syndrome-associated birth defects [1–4]. The variance seen in infant exam parameters in the neonatal period may not represent the variance present later in infancy, so researchers should confirm their own effect sizes from their own study animals when defining significant differences between ZIKV-exposed and control animals. Even with our small sample size and the resulting sample size estimation limitations, these estimations are useful as a starting point for study design and bring the discussion of effect sizes in congenital infection macaque studies to the forefront.

The quantitative infant exam results generate testable hypotheses about specific visual pathway abnormalities in congenitally ZIKV-exposed infants. We hypothesized that the visual pathway may be affected in our ZIKV-exposed infants because it is affected in human infants with congenital ZIKV infection [62]. We wanted to see which parts of the visual pathway may have abnormalities, so we looked for biological connections between quantitative infant exam parameters with similar large effect sizes. Quantitative measurements may be functionally related if they have a similar size of difference between groups, i.e. effect size. We identified multiple parameters with large effect sizes (>1.5) functionally related to visual function. These infant exam parameters include the photoreceptor outer segment layer thickness, electroretinography A wave latency, temporal visual region volume and SNAP orientation at the middle time point. These parameters are functionally related: photoreceptor layer thickness corresponds to photoreceptor function, as measured by the A wave [63], and overall visual function, as assessed by visual orientation tasks in the SNAP orientation construct [30]. Visual function is also impacted by the organization of the temporal visual region, which is involved in object, face and scene perception [64] and alterations in this function may impact visual orientation skills assessed in the SNAP orientation construct. Based on their similar effect sizes and functional relatedness, we hypothesize that a thinner photoreceptor layer results in decreased photoreceptor function, smaller visual temporal visual region, and decreased visual orientation skills. Our hypothesis aligns with the finding that some human children with congenital ZIKV

exposure have a thinned photoreceptor [65]. Abnormalities downstream of the thinned photo-receptor layer remain to be defined in human children, so macaque studies should fill this void by defining specific visual pathway abnormalities in appropriately powered studies. Identification of specific structural abnormalities in young infants that are associated with visual function deficits in childhood could provide a pathway for early diagnosis of neurodevelopmental deficits.

We identified neutrophilic otitis media and bronchopneumonia in ZIKV exposed macaque infants, which is the first time that these acute inflammatory findings have been identified in this model. Congenital ZIKV infection has not been associated with neutrophilic otitis media before, although it is associated with sensorineural hearing loss and rarely with conductive hearing loss [66]. It is unclear whether the otitis media may be within the range of normal findings on histopathologic evaluation of neonatal macaques because the prevalence of otitis media in infant rhesus macaques is unknown. We presume it is a rare finding in macaques because otitis media is found in only 4% of human neonates less than 2 weeks old [67]. We do not know whether the infants with otitis media in our study had hearing loss because the infants with histologic changes did not have ABR testing due to lack of equipment early in our study. Pneumonia is a common cause of death in human infants with congenital ZIKV infection, likely secondary to dysphagia and reflux [68]. The etiology of this acute inflammation may be a bacterial infection, but this cannot be confirmed in all animals because we only obtained bacterial cultures on the lung tissue of the symptomatic infant who had respiratory distress. The functional outcomes of these pathologies included the respiratory distress in one infant; the remaining infant had no respiratory symptoms. Future long-term neurodevelopmental studies of congenital ZIKV exposed macaque infants should include sequential hearing tests on all infants and consider videofluoroscopic swallow studies to better understand the impact of micro aspiration on pulmonary function.

In summary, we demonstrate that moderately size macaque studies are statistically sufficient to define visual, hearing and brain abnormalities, complementing the qualitative clinical and histopathological assessments commonly conducted in ours and other macaque models of prenatal ZIKV exposure [18]. Longitudinal studies of macaque development based on this feasibility study will complement human studies by filling critical knowledge gaps: identification of early neural predictors of neurodevelopmental deficits and pathogenesis of these deficits. Early identification of neurodevelopmental deficits is critical to improving long-term functional outcomes of affected children because early intervention is the only evidence-based intervention that improves outcomes in children with neurodevelopmental deficits [69].

## Supporting information

**S1 Fig. Scoring of representative ERG and VEP waveforms.** These ERG and VEP waveforms were recorded simultaneously from control infant r18076. Red solid arrows: amplitude of labelled waveform features. ERG A- and VEP N2-wave amplitudes are measured relative to pre-flash voltage; ERG B-and VEP P2-wave amplitude measured from preceding A- and N2-waves, respectively (horizontal dotted line). Red dashed arrows: latency-to-peak, all measured from the onset of the flash stimulus (vertical dotted line). The neonatal rhesus photopic ERG contains a prominent short-latency wave (*) prior to the B-wave that decreases in prominence with age and was not quantified for this study.
(TIF)

**S2 Fig. Cortical and subcortical volumetric analysis work flow and parcellations.** (A) Auto-Seg grey matter parcellation with axial (left), sagittal (middle) and coronal (right) views. (B) AutoSeg subcortical structure parcellation with axial (left), sagittal (middle) and coronal

(right) views. Parcellations are overlaid on the subject T1w image, registered to the Emory-Cross 2 week infant template.
(TIF)

**S3 Fig. Maternal viremia and antibody response.** (A) Maternal plasma vRNA loads were measured by qRT-PCR. (B) Estimated EC90 values for serum neutralization of ZIKV assessed prior to infection, 7 dpi, 28 dpi and at maternal necropsy. (C) Estimated EC50 values for serum ZIKV IgG titers obtained prior to infection, 7 dpi, 28 dpi and at maternal necropsy. The dashed line indicates the limit of detection.
(TIF)

**S4 Fig. PRNT and whole virion ELISA dose response curves.**
(TIF)

**S5 Fig. Fetal abdominal circumference, femur length and heart rate measurements.**
(TIF)

**S6 Fig. Infant IgM titers, body fluid viral loads and PRNTs.** Sera from ZIKV-exposed infants were evaluated for the presence of ZIKV-specific IgM by two commercial assays: Abcam (A) and Euroimmun (B). The manufacturer provided positive and negative human IgM controls are shown in grey. Pre and post-ZIKV infection serum collected from an adult rhesus macaque (244667) was used as a species-specific positive and negative control (shown in black). The dashed line indicates the positive cutoff value for each assay. (C) Infant body fluids (plasma, urine, CSF) were assessed for the presence of ZIKV RNA by qRT-PCR at the days indicated and were negative. NA, not applicable (because no samples were available from the stillborn fetus). (D) PRNTs were performed on infant serum from day of life (DOL) 2–5 from liveborn infants and ED90 values were reported in comparison with maternal PRNT ED90 values from necropsy.
(TIF)

**S7 Fig. Infant PRNT dose response curves.** Infant serum was collected at the day of life (DOL) indicated.
(TIF)

**S8 Fig. Localization of ZIKV RNA by ISH in brain tissues from ZIKV-exposed infants.** (A) ZIKV genomic RNA (red dots) in a ZIKV-exposed murine lateral geniculate brain section (20x). ZIKV-exposed fetus/infant brain sections exposed to ZIKV RNA probe (10x) (B) 527421 cerebral cortex (rostral to LGN), (C) 499874 cerebral cortex, (D) 226691 cerebral cortex, (E) 424847 cerebrum containing the lateral geniculate nucleus, (F) 416597 cerebellar brain section.
(TIF)

**S1 Appendix. Effect size and post-hoc sample size calculations SAS code.**
(DOCX)

**S1 Table. Infant ages at examination/evaluation and infant sex.**
(DOCX)

**S2 Table. SNAP constructs.**
(DOCX)

**S3 Table. Infant procedural sedation medication record.**
(DOCX)

**S4 Table. Histopathological description of decidua, placental bed and uterus, and placenta.**
(DOCX)

**S5 Table. Infant clinical course summary.**
(DOCX)

**S6 Table. SNAP effect sizes and sample size estimates.**
(DOCX)

**S7 Table. Ophthalmic exam summary of ZIKV-exposed and control infants.**
(DOCX)

**S8 Table. Ocular coherence tomography effect sizes and sample size estimates.**
(DOCX)

**S9 Table. Visual electrophysiology effects sizes and sample sizes estimates.**
(DOCX)

**S10 Table. Cortical and subcortical brain region volumes with sample size estimates, corrected by total brain volume or intracranial volume.**
(DOCX)

**S11 Table. Fetus/Infant tissue viral loads.**
(DOCX)

**S12 Table. Infant/fetus morphometric measurements at necropsy.**
(DOCX)

**S13 Table. Histopathological description of the lung and middle ear in fetuses/infants.**
(DOCX)

**S14 Table. Histopathological description of fetal/infant tissues excluding the lung and ear.**
(DOCX)

**S15 Table. Summary of sample size estimates for all quantitative infant exams.**
(DOCX)

## Acknowledgments

We thank Nathan Diers, Seth Eaton and Peter Cueno for assistance with OCT segmentation. We thank Clara R. Landucci for making the ear illustration. We thank Marina Emborg for her critical review of the manuscript. We thank Martin Styner for his assistance with the AutoSeg sotware.

## Author Contributions

**Conceptualization:** Michelle R. Koenig, Christina M. Newman, Dawn M. Dudley, Meghan E. Breitbach, Matthew R. Semler, Laurel M. Stewart, Mariel S. Mohns, Maria Dennis, Jennifer M. Hayes, Sallie Permar, Saverio Capuano, III, Thomas C. Friedrich, Thaddeus G. Golos, David H. O'Connor, Emma L. Mohr.

**Data curation:** Elaina Razo, Ann Mitzey, Emma L. Mohr.

**Formal analysis:** Kathryn M. Bach, Michele L. Schotzko, T. Michael Nork, Carol A. Rasmussen, Alex Katz, Jiancheng Hou, Amy Hartman, James Ver Hoeve, Charlene Kim, Mary L. Schneider, Karla Ausderau, Sarah Kohn, Jens Eickhoff, Kathleen M. Antony, Vivek Prabhakaran, Emma L. Mohr.

**Funding acquisition:** Emma L. Mohr.

**Investigation:** Michelle R. Koenig, Ann Mitzey, Christina M. Newman, Dawn M. Dudley, Meghan E. Breitbach, Matthew R. Semler, Laurel M. Stewart, Andrea M. Weiler, Sierra Rybarczyk, Kathryn M. Bach, Mariel S. Mohns, Heather A. Simmons, Andres Mejia, Michael Fritsch, Maria Dennis, Leandro B. C. Teixeira, Michele L. Schotzko, T. Michael Nork, Carol A. Rasmussen, Alex Katz, Veena Nair, Amy Hartman, James Ver Hoeve, Charlene Kim, Mary L. Schneider, Karla Ausderau, Sarah Kohn, Anna S. Jaeger, Matthew T. Aliota, Jennifer M. Hayes, Nancy Schultz-Darken, Kathleen M. Antony, Kevin Noguchi, Xiankun Zeng, Sallie Permar, Emma L. Mohr.

**Methodology:** Emma L. Mohr.

**Resources:** Emma L. Mohr.

**Supervision:** Emma L. Mohr.

**Validation:** Emma L. Mohr.

**Visualization:** Elaina Razo, Ann Mitzey, Emma L. Mohr.

**Writing – original draft:** Michelle R. Koenig, Emma L. Mohr.

**Writing – review & editing:** Michelle R. Koenig, Elaina Razo, Ann Mitzey, Christina M. Newman, Dawn M. Dudley, Thomas C. Friedrich, Thaddeus G. Golos, David H. O'Connor, Emma L. Mohr.

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
