## [Decision Letter · Decision Letter 0]

30 Jul 2020

PONE-D-20-18920

Quantitative definition of neurobehavior, vision, hearing and brain volumes in macaques congenitally exposed to Zika virus

PLOS ONE

Dear Dr. Mohr,

Thank you for submitting your manuscript to PLOS ONE. After careful consideration, we feel that it has merit but does not fully meet PLOS ONE’s publication criteria as it currently stands. Therefore, we invite you to submit a revised version of the manuscript that addresses the points raised during the review process.

Academic editor comments:

First of all, my sincere apologies on the delay on the review process. The subject of the manuscript certainly merits an in-depth investigation. Although the manuscript is interesting and may provide a significant contribution to the literature, there are still some methodological concerns raised by the reviewers that must be clarified before the paper is further considered for publication.

We look forward to receiving your revised manuscript.

Kind regards,

Rafael da Costa Monsanto, M.D.

Academic Editor

PLOS ONE

Journal Requirements:

2.Thank you for stating the following in the Competing Interests section:

[DHO is a paid consultant for Battelle, devoted to research in the areas of assisting in the design and interpretation of their nonhuman primate ZIKV studies. His relationship does not carry with it any restrictions on publication, and any associated intellectual property will be disclosed and processed according to UW-Madison policy. None of the animals used in this study are involved in any studies with Battelle.].

Reviewers' comments:

Reviewer's Responses to Questions

**Comments to the Author**

1. Is the manuscript technically sound, and do the data support the conclusions?

Reviewer #1: Yes

Reviewer #2: Yes

Reviewer #3: Partly

2. Has the statistical analysis been performed appropriately and rigorously? 

Reviewer #1: Yes

Reviewer #2: Yes

Reviewer #3: No

3. Have the authors made all data underlying the findings in their manuscript fully available?

Reviewer #1: Yes

Reviewer #2: Yes

Reviewer #3: Yes

4. Is the manuscript presented in an intelligible fashion and written in standard English?

Reviewer #1: Yes

Reviewer #2: Yes

Reviewer #3: Yes

5. Review Comments to the Author

Reviewer #1: Thank you very much for the opportunity to review the paper titled: "Quantitative definition of neurobehavior, vision, hearing and brain volumes in macaques congenitally exposed to Zika virus". This a dense, but very interesting study on Zika virus infection and brings a lot of knowledge on the different fields. The authors assessed the methodological and statistical feasibility of a congenital ZIKV exposure macaque model for identifying brain abnormalities and also other sensitive organs (eye and ear) that may underlie neurodevelopmental deficits.

The text is very well writen and the language is clear. I congratulate the authors for this study!

I just have few comments regarding this study on its present format.

1. Introduction: I would write the full term before abbreviation (ZIKV).

2. In Materials and Methods: the authors wrote: "Indian-origin rhesus macaques (Macaca mulatta) were inoculated with ZIKV or phosphate buffered saline (PBS)." It is not clear if the PBS should act as a control or the ZIKV was inoculated WITH PBS. What is the idea of using PBS? Please clarify.

3. What was assessed in indirect ophthalmoscopy? Please describe the parameters.

4. Why the authors decided to observe wave IV (superior olivary complex) behavior instead of wave III (cochlear nucleus) or maybe V (inferior coliculus and lateral lemniscus)?

5. In references, please use the complete references format according to what is described in PlosONE webpage.

Reviewer #2: The article entitled "Quantitative definition of neurobehavior, vision, hearing and brain volumes in macaques congenitally exposed to Zika virus" by Koenig and colleagues describes a quantitative definition to analyse various phenotypes of the Congenital Zika Syndrome in macaques. I find this study could be of interest and the authors could contribute to a better characterisation of developmental events impaired caused by Zika virus infection. However, some key points should be addressed:

1) Microcephaly is the hallmark of Congenital Zika Syndrome. What could account for the lack of this phenotype in your model?

2) It is not clear if the authors managed to produce the Congenital Zika Syndrome model with ZIKV vertical transmission.

3) Even without the statistical analysis there could have been an effort to describe qualitatively the various phenotypes.

4) The paper could be re-written as a method paper or a qualitative paper.

Reviewer #3: Synopsis: This manuscript describes detailed evaluation of neonatal rhesus macaques born to dams exposed to Zika virus (n=5) or an uninfected control dam (n=1). In this study, the authors followed the neonates for 8 d post Caesarean birth at 155 dGA, followed by necropsy. Assessments of neurobehavior, visual and hearing function, as well as brain MRI were conducted during the time of observation, and histopathology post-necropsy. In general, the sample size was too small to reveal significant statistical differences in any of the observed parameters with the control animal. This is not surprising, as previous studies indicate a broad spectrum of phenotypes in the Rhesus macaque congenital infection model. This model has also been shown to result in pregnancy loss, which occurred in one of this studies’ animals. Nevertheless, a strength of this study is the wide range of assays deployed to evaluate ZIKV-associated phenotypes, potentially setting the standard for future studies.

Specific comments/ concerns:

1. The post-hoc power analysis has the potential to add significant value to this study by contributing information to future experimental design. However, it is unclear if it is valid to calculate and effect size based on the single control animal, since no standard deviation can be calculated from the control group. Would it be preferable to treat all animals as a single population to calculate SD? Such results could still be extrapolated to a power analysis.

2. Has the NBAS or similar test been applied to human neonates post in utero ZIKV exposure? If so, is an effect of viral infection detected early after birth, or, as mentioned in the introduction (lines 63-64), are abnormalities observed later? Could such results suggest a time post-birth in the NHP model that subtle phenotypes due to in utero infection might be observed.

3. Minor point: In figure 1, the figure has the abbreviation NBAS, while the legend uses the abbreviation SNAP.

4. Minor point: Could the color for the control animal be changed to bright green or something else that makes it more easily identified in the figures?

6. PLOS authors have the option to publish the peer review history of their article (what does this mean?). If published, this will include your full peer review and any attached files.

Reviewer #1: No

Reviewer #2: No

Reviewer #3: No

---

## [Author Response · Author response to Decision Letter 0]

15 Sep 2020

Response to reviewers

Reviewer #1: Thank you very much for the opportunity to review the paper titled: "Quantitative definition of neurobehavior, vision, hearing and brain volumes in macaques congenitally exposed to Zika virus". This a dense, but very interesting study on Zika virus infection and brings a lot of knowledge on the different fields. The authors assessed the methodological and statistical feasibility of a congenital ZIKV exposure macaque model for identifying brain abnormalities and also other sensitive organs (eye and ear) that may underlie neurodevelopmental deficits.

The text is very well written and the language is clear. I congratulate the authors for this study!

I just have few comments regarding this study on its present format.

We appreciate this reviewer’s positive response about our manuscript languate and style. 

1. Introduction: I would write the full term before abbreviation (ZIKV).

This change has been made as recommended.

2. In Materials and Methods: the authors wrote: "Indian-origin rhesus macaques (Macaca mulatta) were inoculated with ZIKV or phosphate buffered saline (PBS)." It is not clear if the PBS should act as a control or the ZIKV was inoculated WITH PBS. What is the idea of using PBS? Please clarify.

The single control pregnancy was inoculated with phosphate buffered saline to mirror the same amount of prenatal stress associated with inoculation as the ZIKV-challenged dams experienced. I clarified the language in the methods section to say that “… were inoculated with ZIKV alone or phosphate buffered saline (PBS) alone during the first trimester… “. The first sentence of the Results section explains the inoculation substances as well: “Five dams were inoculated with ZIKV and one dam was inoculated with PBS in the first trimester”. 

3. What was assessed in indirect ophthalmoscopy? Please describe the parameters.

Indirect ophthalmoscopy assessed the parameters described in Supplementary Table 7, which includes the pupillary response, eyelids and adnexa, conjunctiva, cornea, anterior chamber, iris, lenses, vitreous, hyaloid artery, tunica vasculosa lentis, optic nerve, retina, and choroid. I described the finding of a corneal defect and retinal pigmented epithelium mottling in the Results section, and highlighted that these results were identified with indirect ophthalmoscopy in this revision. The revised sentence reads “Two of the ZIKV-exposed infants had minor ocular defects similar to defects observed in human infants with congenital ZIKV infection identified with indirect ophthalmoscopy, a corneal defect and retinal pigmented epithelium mottling (S7 table). No other ocular birth defects associated with congenital Zika syndrome were identified [1], including no optic nerve hypoplasia, choroidal lesions, lens abnormalities or vitreous opacities.”

4. Why the authors decided to observe wave IV (superior olivary complex) behavior instead of wave III (cochlear nucleus) or maybe V (inferior coliculus and lateral lemniscus)?

In humans, wave V is examined when assessing hearing thresholds because it is the most robust and reliable out of the five waveforms. Waves I-III are not typically looked at for this reason. Brainstem auditory evoked potentials in rhesus macaques have only 4 main vertex-positive peaks in response to click sounds, which is one less than humans. Studies comparing human and rhesus BAEP hypothesize that the human peak I and II combine into a single peak in rhesus because of their shorter auditory nerve [2]. Peak II in the rhesus macaque corresponds to peak III in humans, peak III to peak IV, and peak IV to peak V. Due to the finding that rhesus macaques only have 4 waveforms, and that wave IV corresponds to the inferior colliculus response in macaques, we utilize wave IV in data analysis and threshold estimations.

5. In references, please use the complete references format according to what is described in PlosONE webpage.

I have updated the references as recommended.

Reviewer #2: The article entitled "Quantitative definition of neurobehavior, vision, hearing and brain volumes in macaques congenitally exposed to Zika virus" by Koenig and colleagues describes a quantitative definition to analyse various phenotypes of the Congenital Zika Syndrome in macaques. I find this study could be of interest and the authors could contribute to a better characterisation of developmental events impaired caused by Zika virus infection. However, some key points should be addressed:

1) Microcephaly is the hallmark of Congenital Zika Syndrome. What could account for the lack of this phenotype in your model?

Microcephaly is observed in only 5% of infants born to women with laboratory evidence of Zika virus infection during pregnancy [3]. Up to 15% of infants born to women with laboratory evidence of Zika virus infection during pregnancy have a Zika-associated birth defect [3]. It is unlikely that we will observe microcephaly and severe birth defects in our macaque model with the small sample size of four liveborn infant macaques. In future studies we will focus on the most common phenotype of congenital Zika virus infection – developmental delays, which occurs in about 30% of children. We did not focus on these long-term developmental outcomes in this study because we assessed tissue viral loads at 1 week of age. 

2) It is not clear if the authors managed to produce the Congenital Zika Syndrome model with ZIKV vertical transmission.

We did not observe microcephaly or severe birth defects such contractures, qualitatively apparent brain anomalies, chorioretinal anomalies or hearing loss. These severe birth defects occur in up to 15% of infants born to women with laboratory evidence of Zika virus infection during pregnancy [3]. We were not sure when we began these studies if we would observe congenital Zika syndrome, so set out to capture as much quantitative detail about vision, hearing and brain volumes as we could, to define more subtle differences. 

3) Even without the statistical analysis there could have been an effort to describe qualitatively the various phenotypes.

We did not observe a phenotype consistent with severe Zika-associated birth defects, which means there is not a “birth defect” phenotype to describe. Subtle brain abnormality phenotypes, such as alterations in visual electrophysiology or brain region volume changes, require sufficient number of control animals. We do not have a sufficient number of animals for these types of studies in our neonatal macaques because we decided to change our focus to long-term neurodevelopment, as these are a more common phenotype (28%) of congenital Zika exposure [4], and because neurodevelopment is best studied longitudinally in childhood, rather than just the first week of life. This current macaque study was carefully designed to describe birth defect phenotypes and define viral tissue tropism, something that has to be done in the early neonatal period. We agree with the reviewer that future studies should define phenotypes and we hope to do this with long-term studies that describe subtle neurodevelopmental deficits along with the underlying neuropathway abnormalities.

4) The paper could be re-written as a method paper or a qualitative paper.

We appreciate this comment because one of our goals was to describe a comprehensive battery of tests that need be done in macaque models of congenital Zika infection in order to fully define the effect of therapeutic interventions, such as vaccines or antiviral therapy. Much of the field until this point has utilized endpoints such as microcephaly or tissue viral loads as the endpoint of therapeutic interventions. These endpoints tell only part of the story and leave out the more common outcome, developmental deficits and subtle brain abnormalities that underlie these deficits. We hope that our detailed methods section describing optical coherence tomography, visual electrophysiology, hearing tests, and brain region volume analyses in infant macaques will pave the way for other groups to perform these quantitative tests in their macaque models of congenital viral infections. We took our manuscript a step beyond the methods description to help researchers design future studies of congenital viral infection interventions by performing post-hoc sample size calculations. Hopefully our methods descriptions and sample size estimates will assist other groups in their studies of congenital viral infection in a macaque model.

Reviewer #3: Synopsis: This manuscript describes detailed evaluation of neonatal rhesus macaques born to dams exposed to Zika virus (n=5) or an uninfected control dam (n=1). In this study, the authors followed the neonates for 8 d post Caesarean birth at 155 dGA, followed by necropsy. Assessments of neurobehavior, visual and hearing function, as well as brain MRI were conducted during the time of observation, and histopathology post-necropsy. In general, the sample size was too small to reveal significant statistical differences in any of the observed parameters with the control animal. This is not surprising, as previous studies indicate a broad spectrum of phenotypes in the Rhesus macaque congenital infection model. This model has also been shown to result in pregnancy loss, which occurred in one of this studies’ animals. Nevertheless, a strength of this study is the wide range of assays deployed to evaluate ZIKV-associated phenotypes, potentially setting the standard for future studies.

Specific comments/ concerns:

1. The post-hoc power analysis has the potential to add significant value to this study by contributing information to future experimental design. However, it is unclear if it is valid to calculate and effect size based on the single control animal, since no standard deviation can be calculated from the control group. Would it be preferable to treat all animals as a single population to calculate SD? Such results could still be extrapolated to a power analysis.

We appreciate this comment because it allowed us to clarify our statistical methods description. We used the standard deviation estimation of the ZIKV group to estimate the effect sizes. The underlying assumption is that there is no difference in variability in the measurements between groups. This is similar to the calculation used for Glass's delta effect sizes where the standard deviation of only the control group is used for the calculation of the effect sizes. We have clarified this in the Statistical Analysis section of the revised manuscript. Alternatively, standard deviations of the pooled groups could be used. However, this could result into inflated standard deviations estimates if there are large differences in the group means. We added a sentence in the Statistical Analyses sections: “Due to the small sample size in the control group, only the standard deviations of the ZIKV-exposed group observations were used for the effect size estimation.”

2. Has the NBAS or similar test been applied to human neonates post in utero ZIKV exposure? If so, is an effect of viral infection detected early after birth, or, as mentioned in the introduction (lines 63-64), are abnormalities observed later? Could such results suggest a time post-birth in the NHP model that subtle phenotypes due to in utero infection might be observed.

I appreciate this interesting question from the reviewer. The question is whether the newborn behavioral test we utilized has been used in human development studies following congenital viral infection, and if so, is there is an optimal time during childhood when we would expect to find developmental abnormalities. First, the neurobehavioral assessment we used in newborn macaques is the Schneider Neonatal Assessment for Primates (SNAP) [5-9], which is based on the human newborn assessment called the Brazelton Newborn Behavioral Assessment Scale [10]. Human congenital infection researchers use the Brazelton Newborn Behavioral Assessment to define infant development after prenatal CMV exposure [11], but its use in infants with congenital Zika exposure has not been described. We selected this test because it has been well validated in macaques specifically in the neonatal age we were evaluating. The SNAP is a well known test for neonatal macaques and has been used by other researchers to define infant macaque neurobehavior in postnatal Zika virus infection [12]. Human studies of infants with congenital Zika virus exposure indicate that developmental deficits become apparent in early childhood, by about 24 months. Macaques develop faster than humans, so we hypothesize that we’d start seeing developmental deficits by 8 months of age in macaques. We will do these long-term studies in the future now that we know they are methologically and statistically feasible from this foundational study.

3. Minor point: In figure 1, the figure has the abbreviation NBAS, while the legend uses the abbreviation SNAP.

We have made this change as suggested.

4. Minor point: Could the color for the control animal be changed to bright green or something else that makes it more easily identified in the figures?

We selected the color grey to avoid color differentiation challenges for persons with red-green color blindness. We think the grey color is still apparent and nicely contrasts the ZIKV-exposed infants in multiple colors with the control infant in the non-color of grey. We hope this reviewer will understand the logic of selecting this color for the control infant.

Reviewer #4: In this paper, the authors plan to assess the methodological and statistical feasibility of a congenital ZIKV exposure macaque model for identifying infant neurobehavior and brain abnormalities that may underlie neurodevelopmental deficits

Gap presented: “no established nonhuman primate model for defining the neuropathogenesis of the most common phenotype of congenital ZIKV infection: children who develop neurodevelopmental deficits but lack the birth defects found in congenital Zika syndrome”

This is an appropriate experimental model to attempt to create. However, most of the studies that have found neurobehavioral problems in children with a history of in utero Zika virus exposure have found largest differences in neurocognitive testing is performed when at least 12-18 months old (Faiçal, A. V., et al. (2019). (Neurodevelopmental delay in normocephalic children with in utero exposure to Zika virus." BMJ Paediatr Open 3(1): e000486; Rice, M. E., et al. (2018); Vital Signs: Zika-Associated Birth Defects and Neurodevelopmental Abnormalities Possibly Associated with Congenital Zika Virus Infection - U.S. Territories and Freely Associated States, 2018." MMWR. Morbidity and mortality weekly report 67(31): 858-867.). As the authors allude to in the paper, the infant neurobehavior metrics will be best evaluated by performing the non-invasive studies proposed in this paper soon after birth and then following up with neurocognitive testing in toddler-equivalent age macaques.

We wholeheartedly agree with this reviewer that non-invasive studies performed in infancy followed by neurocognitive testing in toddler-equivalent age macaques will be the best way to identify early predictors of deficits, but in order to carry out these long-term studies successfully, we needed to demonstrate that these noninvasive studies are methodogically and statistically feasible. We describe in our introduction why a feasibility study is necessary to ensure a robust and statistically sound experimental design in long-term studies, much like how human clinical trials often begin with a feasibility study. “None of these [previous ZIKV] studies have defined long-term neurodevelopmental deficits or outlined a clear study design for how to evaluate long-term neurodevelopment in congenital ZIKV-exposed infant macaques. Such a study design needs to be well planned, with proven quantitative neurodevelopmental outcomes and sufficient sample sizes. Before long-term studies defining the pathogenesis of neurodevelopmental deficits in ZIKV-exposed macaques are undertaken, we must determine whether tests defining neurodevelopmental outcomes, such as quantitative structural brain imaging, ocular examinations and hearing tests, are feasible in infant macaques.”

We hope this reviewer will agree that performing a feasibility study to refine methods and estimate sample sizes is necessary before embarking on the expensive, long-term studies necessary for defining the pathogenesis of developmental deficits. Nonhuman primates are a precious scientific resource, especially now in the time when SARS-CoV-2 vaccine and therapeutic trials are involving significant numbers of nonhuman primates. We want to ensure that future long-term studies are optimally designed and powered to capture all necessary data points. 

References for this Response to Reviewers

1. Yepez JB, Murati FA, Pettito M, Penaranda CF, de Yepez J, Maestre G, et al. Ophthalmic Manifestations of Congenital Zika Syndrome in Colombia and Venezuela. JAMA ophthalmology. 2017;135(5):440-5. Epub 2017/04/19. doi: 10.1001/jamaophthalmol.2017.0561. PubMed PMID: 28418539.

2. Møller AR, Burgess J. Neural generators of the brain-stem auditory evoked potentials (BAEPs) in the rhesus monkey. Electroencephalography and clinical neurophysiology. 1986;65(5):361-72. Epub 1986/09/01. doi: 10.1016/0168-5597(86)90015-8. PubMed PMID: 2427327.

3. Rice ME, Galang RR, Roth NM, Ellington SR, Moore CA, Valencia-Prado M, et al. Vital Signs: Zika-Associated Birth Defects and Neurodevelopmental Abnormalities Possibly Associated with Congenital Zika Virus Infection - U.S. Territories and Freely Associated States, 2018. MMWR Morbidity and mortality weekly report. 2018;67(31):858-67. Epub 2018/08/10. doi: 10.15585/mmwr.mm6731e1. PubMed PMID: 30091967; PubMed Central PMCID: PMCPMC6089332.

4. Mulkey SB, Arroyave-Wessel M, Peyton C, Bulas DI, Fourzali Y, Jiang J, et al. Neurodevelopmental Abnormalities in Children With In Utero Zika Virus Exposure Without Congenital Zika Syndrome. JAMA pediatrics. 2020. Epub 2020/01/07. doi: 10.1001/jamapediatrics.2019.5204. PubMed PMID: 31904798.

5. Laughlin NK, Lasky RE, Giles NL, Luck ML. Lead effects on neurobehavioral development in the neonatal rhesus monkey (Macaca mulatta). Neurotoxicology and teratology. 1999;21(6):627-38. Epub 1999/11/24. PubMed PMID: 10560769.

6. Schneider ML, Moore CF, Kraemer GW, Roberts AD, DeJesus OT. The impact of prenatal stress, fetal alcohol exposure, or both on development: perspectives from a primate model. Psychoneuroendocrinology. 2002;27(1-2):285-98. Epub 2001/12/26. PubMed PMID: 11750784.

7. Schneider ML, Roughton EC, Koehler AJ, Lubach GR. Growth and development following prenatal stress exposure in primates: an examination of ontogenetic vulnerability. Child development. 1999;70(2):263-74. Epub 1999/04/28. PubMed PMID: 10218255.

8. Schneider ML, Suomi SJ. Neurobehavioral Assessment in Rhesus Monkey Neonates (Macaca mulatta): Developmental Changes, Behavioral Stability and Early Experience. Infant Behavior and Development. 1992;15:155-77. PubMed PMID: 1002; PubMed Central PMCID: PMC1002.

9. Coe CL, Lubach GR, Crispen HR, Shirtcliff EA, Schneider ML. Challenges to maternal wellbeing during pregnancy impact temperament, attention, and neuromotor responses in the infant rhesus monkey. Developmental psychobiology. 2010;52(7):625-37. Epub 2010/10/01. doi: 10.1002/dev.20489. PubMed PMID: 20882585; PubMed Central PMCID: PMCPMC3065369.

10. Brazelton TB. Assessment of the infant at risk. Clinical obstetrics and gynecology. 1973;16(1):361-75. Epub 1973/03/01. PubMed PMID: 4575534.

11. Ancora G, Lanari M, Lazzarotto T, Venturi V, Tridapalli E, Sandri F, et al. Cranial ultrasound scanning and prediction of outcome in newborns with congenital cytomegalovirus infection. The Journal of pediatrics. 2007;150(2):157-61. Epub 2007/01/24. doi: 10.1016/j.jpeds.2006.11.032. PubMed PMID: 17236893.

12. Maness NJ, Schouest B, Singapuri A, Dennis M, Gilbert MH, Bohm RP, et al. Postnatal Zika virus infection of nonhuman primate infants born to mothers infected with homologous Brazilian Zika virus. Scientific reports. 2019;9(1):12802. Epub 2019/09/07. doi: 10.1038/s41598-019-49209-7. PubMed PMID: 31488856.

---

## [Decision Letter · Decision Letter 1]

6 Oct 2020

Quantitative definition of neurobehavior, vision, hearing and brain volumes in macaques congenitally exposed to Zika virus

PONE-D-20-18920R1

Dear Dr. Mohr,

We’re pleased to inform you that your manuscript has been judged scientifically suitable for publication and will be formally accepted for publication once it meets all outstanding technical requirements.

Kind regards,

Rafael da Costa Monsanto, M.D.

Academic Editor

PLOS ONE

Additional Editor Comments (optional):

Reviewers' comments:

Reviewer's Responses to Questions

**Comments to the Author**

1. If the authors have adequately addressed your comments raised in a previous round of review and you feel that this manuscript is now acceptable for publication, you may indicate that here to bypass the “Comments to the Author” section, enter your conflict of interest statement in the “Confidential to Editor” section, and submit your "Accept" recommendation.

Reviewer #1: All comments have been addressed

Reviewer #3: All comments have been addressed

2. Is the manuscript technically sound, and do the data support the conclusions?

Reviewer #1: Yes

Reviewer #3: Yes

3. Has the statistical analysis been performed appropriately and rigorously? 

Reviewer #1: Yes

Reviewer #3: Yes

4. Have the authors made all data underlying the findings in their manuscript fully available?

Reviewer #1: Yes

Reviewer #3: Yes

5. Is the manuscript presented in an intelligible fashion and written in standard English?

Reviewer #1: Yes

Reviewer #3: Yes

6. Review Comments to the Author

Reviewer #1: (No Response)

Reviewer #3: (No Response)

7. PLOS authors have the option to publish the peer review history of their article (what does this mean?). If published, this will include your full peer review and any attached files.

Reviewer #1: No

Reviewer #3: No

---

## [Editor Report · Acceptance letter]

12 Oct 2020

PONE-D-20-18920R1 

Quantitative definition of neurobehavior, vision, hearing and brain volumes in macaques congenitally exposed to Zika virus 

Dear Dr. Mohr:

I'm pleased to inform you that your manuscript has been deemed suitable for publication in PLOS ONE. Congratulations! Your manuscript is now with our production department. 

Kind regards, 

on behalf of

Dr. Rafael da Costa Monsanto 

Academic Editor

PLOS ONE